

# Understanding the drivers of near-surface winds in Adélie land, East Antarctica

Cécile Davrinche[1], Anaïs Orsi[1,2], Cécile Agosta[1], Charles Amory[1,3], and Christoph Kittel[3]

[1]Laboratoire de Sciences du Climat et de l'Environnement, LSCE-IPSL, CEA, CNRS, UVSQ, UMR8212, Université Paris Saclay, Gif-sur-Yvette, France
[2]Department of Earth, Ocean and Atmospheric Sciences, The University of British Columbia, Vancouver, BC, Canada
[3]Institut des Géosciences de l'Environnement (IGE), Université Grenoble Alpes/CNRS/IRD/G-INP, Grenoble, France

**Correspondence:** Cécile Davrinche, cecile.davrinche@lsce.ipsl.fr

**Abstract.** Near-surface winds play a crucial role in the climate of Antarctica, but accurately quantifying and understanding their drivers is complex. They result from the contribution of two distinct families of drivers: large-scale pressure gradient, and surface-induced pressure gradients known as katabatic and thermal wind. The extrapolation of vertical potential temperature above the boundary layer down to the surface enables us to separate and quantify the contribution of these different pressure

gradients in the momentum budget equations. Using this method applied to outputs of the regional atmospheric model MAR at a 3-hourly resolution, we find that the seasonal and spatial variability of near-surface winds in Adélie Land is dominated by surface processes. On the other hand, high temporal variability (3-hourly) is mainly controlled by large-scale variability everywhere in Antarctica, except in the coastal area. In these coastal regions, although the katabatic acceleration surpasses all other accelerations in magnitude, none of the katabatic nor large-scale accelerations can be identified as primary drivers

of near-surface winds variability. Strong wind speed events in coastal Antarctica are driven by both katabatic and large-scale accelerations, as well as the angle between them.

## 1 Introduction

Near-surface winds play a key role in the Antarctic climate system. First, they contribute to an active mass exchange between the continent and sub-polar latitudes. They transport cold surface air northward, which causes warmer subpolar latitudes air masses to rise and travel northward to replenish the cold air removed (Parish and Bromwich, 1998). Moreover, they have a

major influence on the ice sheet surface mass balance. At the surface, they redistribute surface snow across the continent, which can sublimate during transport in the lower atmosphere (Lenaerts et al., 2012; Amory et al., 2021; Gerber et al., 2023). Additionally, high near-surface wind speeds enhance the mass and energy exchange at the surface-atmosphere interface and contribute to increase sublimation of surface snow (Bintanja, 1998). Furthermore, near-surface winds originating from the cold and dry inner continent supply the lower troposphere with unsaturated air as they flow downslope and adiabatically warm

up (Gallée and Pettré, 1998). This causes precipitating snow to sublimate into the atmosphere (Vignon et al., 2019; Jullien et al., 2020) and thus decreases the amount of precipitation reaching the ground up to 35 % on the margins of East Antarctica (Grazioli et al., 2017).



These winds are complex, because they result from two different families of drivers: in the free atmosphere, winds are solely governed by large-scale pressure gradients. Additionally, in the boundary layer, the dense, cold surface air, caused by surface net radiative cooling (followed by turbulent sensible heat exchange between the atmosphere and the surface) is accelerated by gravity on the steep surface slope, generating a divergent flow called katabatic wind (Gallée and Pettré, 1998). At the same time, the accumulation of cold air over the lowest part of the slope and the sea ice induces a poleward flow, the thermal wind, which opposes the katabatic flow near the foot of the slope (Vihma et al., 2011).

It is important to disentangle the impact of large-scale and boundary layer forcings on Antarctic near-surface winds, because they have different drivers and might evolve differently in the future. In the next decades, during winter, large-scale forcing is expected to weaken at the ice sheet ocean margins due to a more positive southern annular mode (SAM) (Hazel and Stewart, 2019; Neme et al., 2022).

Simultaneously, katabatic forcing could also decrease in a warmer climate due to the increase in downward longwave radiation. However, the decrease in boundary layer stability might also induce stronger mixing with upper geostrophic winds by increased vertical momentum transfer (Bintanja et al., 2014). The resulting change in wind speed is thus very uncertain and depends greatly on the region of Antarctica, with potential cancellation between regions of increase and decrease (Bracegirdle et al., 2008).

In order to study the temporal variability of Antarctic near-surface winds, it is thus essential to look at each component of the momentum budget separately. In previous studies, the katabatic nature of Antarctic near-surface winds forcing diagnosed using the directional constancy has been overemphasised. It had been suggested that the katabatic nature of winds could be estimated using Weibull shape parameters (Sanz Rodrigo et al., 2013) like in Greenland (Gorter et al., 2014). However, in Antarctica, the large-scale pressure gradient is also directed from the interior to the coast, which has led to an overestimation of the role of the katabatic forcing for decades (Parish and Cassano, 2003). Instead, a full decomposition of the momentum budget with separation of large-scale and boundary layer contributions is necessary.

The momentum budget decomposition has proven to be a useful tool to study the spatial variability of the different acceleration terms for modelled monthly averaged wind fields ((van den Broeke and van Lipzig, 2003; Parish and Cassano, 2001)). Fewer studies have focused on understanding the inter-diurnal variability of these winds. Yasunari and Kodama (1993) tackled this aspect, albeit at a 30 m level and focusing only on periods ranging from 30 to 60 days. Unfortunately, this range excludes the analysis of short events such as high wind speed events which typically last for less than two days. Renfrew and Anderson (2002) conducted case studies at a 3-hourly resolution using AWS data but had to assume katabatic nature of winds due to the lack of vertical depiction of the atmosphere.

Here we identify the drivers of the temporal variability at a regional scale in East Antarctic near-surface winds by computing the momentum budget in the atmosphere. Compared to previous approaches, our study focuses on understanding the variability of the near-surface winds (7 m above ground level) for a larger range of time-scales using a more accurate diagnosis obtained through an extensive analysis of the vertical profiles of the atmosphere. We first quantify the dominant components of the momentum budget by analysing their spatial and seasonal variations. Then, we focus on the correlations between the different acceleration terms and the total wind speed at a 3-hourly resolution.





## 2 Data and methods

### 2.1 Data

#### 2.1.1 Field observations over a transect in Adélie Land

We focus on the East Antarctic region located between coastal Adélie Land and the Antarctic Plateau (Fig. 1), taking advantage of the supply route between Dumont d'Urville station (DDU, 66.7 °S; 139.8 °E, 0 m above sea level and Concordia station, Dome C (DC, 75.1 °S; 123.3 °E, 3233 m asl). This transect is typical of the climatology of the region, with downslope flow from the East Antarctic Plateau to the coast, and fast easterlies on the shore. Coastal Adélie Land is known for its very strong near-surface winds, with the highest wind speed recorded in Antarctica (96 m s$^{-1}$) monitored at DDU in the late 70's (Wendler, 1990), which makes it an ideal area to study the drivers of near-surface wind variability.

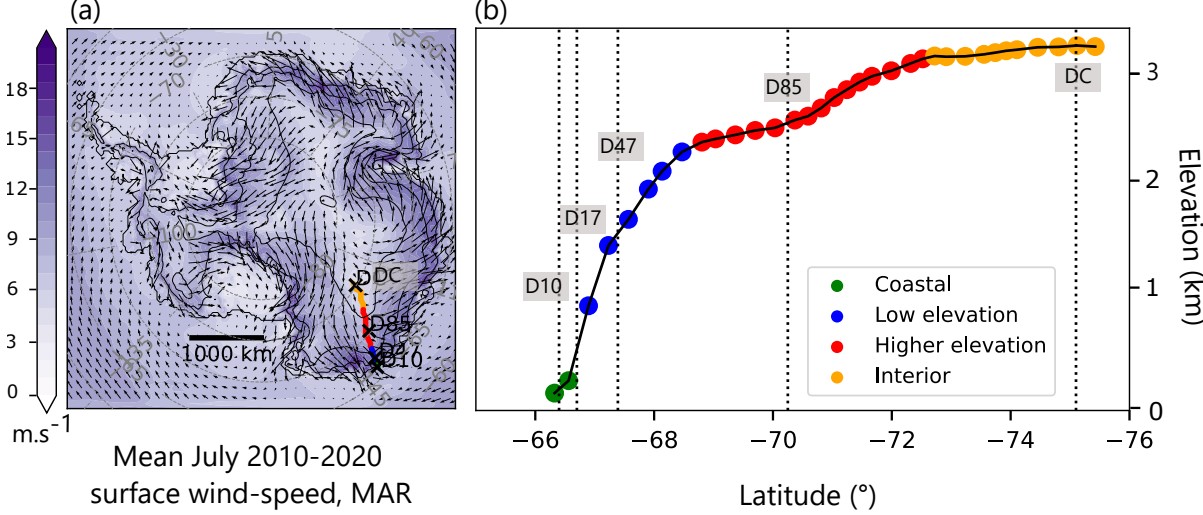

**Figure 1.** (a) Map of average July 2010-2020 norm of near-surface wind speed (MAR). Superimposed are the mean vectors. Black solid lines are for elevations contours every 500 m (asl). The transect is indicated in coloured dots. Four weather stations are indicated: D10, D47, D85 and Dome C (DC). Dumont d'Urville (DDU) is located 5 km offshore of D10, 34 km of D17. (b) Elevation profile along the transect extracted on the 35-km MAR grid. For both plots, color dots represent the different sectors detailed at Table 2, with blue dots on coastal area, red dots on lower elevation, orange dots on high elevations, and green dots on the Antarctic plateau.

This supply route is well instrumented, with six weather stations described in Table 1 and shown on Fig. 1. These stations record wind speed. It enables us to assess the model ability to represent the winds, in a wide range of conditions. Four Automatic Weather Stations (AWS) record temperature and wind speed at approximately 2 m above ground level (agl), with data provided at a 3-hourly resolution . Additionally we use 3-hourly quality-controlled wind speed from two weather-profiling towers: a 7-m tower at D17 (Amory et al., 2017, D17, CALVA project) and the 45-m «American tower» at DC (Genthon et al., 2021). These data are slightly better correlated to our model, therefore, when available, we will rather compare our model to the towers. All



these observations are available even during winter-time, when wind speeds are particularly high (seasonal maximum) and the diurnal cycle is very weak (polar night), leading to favourable katabatic conditions. Therefore, we focus more specifically on the months of July 2010-2020.

**Table 1.** Weather stations located along the transect.

| Station name | Type | Lon. (°E) | Lat. (°N) | Elevation (m asl) | Mean wind speed (m s$^{-1}$) | Period |
|---|---|---|---|---|---|---|
| D10 | AWS | 139.8 | -66.7 | 243 | 6.6 | 2017-2021 |
| D17 | 7-m tower | 139.8 | -66.7 | 438 | 9.7 | 2010-2018 |
| D47 | AWS | 138.7 | -67.4 | 2008 | 12.2 | 2012-2021 |
| D85 | AWS | 134.1 | -70.3 | 2624 | 6.4 | 2017-2018 |
| DC-aws | AWS | 123.3 | -75.1 | 3265 | 3.5 | 2012-2015, 2017-2021 |
| DC-tower | 45-m tower | | | | 3.8 | 2009-2019 |

### 2.1.2 Regional atmospheric model

Our goal is to disentangle the contribution of large-scale and boundary-layer drivers in shaping the near-surface winds of Antarctica. In order to do this, we need a description of the vertical atmospheric column, which is only available in radiosoundings at the two extremities of our transect, DDU and DC. Consequently, due to the scarcity of observations, we perform our study using outputs from the regional atmospheric model MAR v3.11 on the period 2010-2020 (https://gitlab.com/Mar-Group/MARv3), after evaluation of this model for near-surface winds (Section 3.1). MAR is a regional hydrostatic model that takes into account specific physical properties of the Antarctic region, in particular a multi-layer snow model based on CROCUS (Brun et al., 1992; Vionnet et al., 2012), with several adaptations for Antarctica, including meltwater refreezing and parametrized fresh snow density (Agosta et al., 2019). Topography, ice mask and rock mask are derived from Fretwell et al. (2013). The model bases are extensively described in Gallée and Schayes (1994), and a description of the adaptation of MAR to the Antarctic ice sheet can be found in Agosta et al. (2019) and Kittel et al. (2021). Relative to previous studies over the Antarctic ice sheet (Agosta et al., 2019), the version used in this study improves the cloud lifetime, the model stability and its computational efficiency, and the inclusion of rock outcrops, as in Mottram et al. (2020) and Kittel et al. (2021). In addition, MARv3.11 includes a correction of the cloud microphysics in the upper relaxation zone, where clouds were set to zero in previous versions of the model (Kittel et al., 2021). We increased the snow albedo by 5 % (relative to the previous value) in agreement with recent model evaluation performed at DC.

We use 3-hourly outputs of MARv3.11 with 24 vertical atmospheric levels (first model level ∼2 m agl), 30 snow/ice layers with a fixed 20 m thickness, and an horizontal resolution of 35 km. MAR is forced with 6-hourly outputs of the ERA5 reanalysis (Hersbach et al., 2020) at its lateral boundaries (temperature, wind, humidity) and for upper-air relaxation at the top of the troposphere (temperature, wind), and with daily outputs at the surface of the ocean (sea surface temperature, sea ice concentration).





### 2.1.3 Coast-to-plateau transect on the model grid

The spatial variability of near-surface winds is strongly linked to the topography of Antarctica with the strongest winds just
under the steepest slopes. The supply route between DDU and DC crosses a wide range of slopes which enables us to study
the various wind drivers, in particular the katabatic acceleration. On the 35-km MAR grid, we extract the DDU-DC transect
by following the steepest-slope trajectory upstream and downstream of D47. This transect reaches an upstream location close
to DC and a downstream location close to DDU station (Fig. 1). We divide the transect into four elevation bins with different
slopes, similar to van den Broeke et al. (2002), which are detailed in Table 2 and shown in Fig. 1: a coastal region at the foot
of the slope (0-100 m asl), a low elevation region with steep slopes (100-2300 m asl), a higher elevation region with gentler
slopes (2300-3100 m asl) and the nearly flat plateau (3100-3300 m asl). By construction, the transect follows the steepest slope
direction, which enables us to capture the spatial variability of wind, from its formation on the plateau, to its acceleration along
the slopes of Adélie land, up to the coastal area.

**Table 2.** Characteristics of regions defined along the study transect on the 35 km MAR grid. Transect location is shown on Fig. 1.

| Section name | Elevation range (m asl) | Average distance to coast (km) | Average slope (m km$^{-1}$) | Nb of grid cells |
|---|---|---|---|---|
| Coastal | 0-100 | 18 | 5.9 | 2 |
| Lower elevation | 100-2300 | 158 | 8.8 | 6 |
| Higher elevation | 2350-3100 | 525 | 1.6 | 15 |
| Plateau | 3100-3300 | 969 | 0.1 | 11 |

## 2.2 Method

### 2.2.1 Separation of large-scale and surface forcings in the vertical potential temperature profile

The goal of this study is to separate the main drivers of near-surface winds variability. near-surface winds are the result of
two types of forcing: the large-scale pressure gradient and the additional pressure gradients associated with the vicinity of
the surface. van den Broeke and van Lipzig (2003) showed that we can separate the pressure gradient force (PGF) into the
contribution of surface and large-scale using the potential temperature. The vertical potential temperature profile in the free
atmosphere (i.e. above the boundary layer) is approximately linear (see Fig. 2). Well above the boundary layer (typically
above 500 hPa), the potential temperature is only influenced by large-scale pressure gradients. Thus, we linearize the potential
temperature above the boundary layer, and extrapolate it to the surface:

$$\theta_0(x,y,z) = \gamma_0(x,y) \cdot z + \tau_0(x,y) \tag{1}$$

with $z$ the altitude agl, $\gamma_0$ the vertical gradient of the background potential temperature in the free atmosphere (in K m$^{-1}$),
and $\tau_0$ the intercept of $\theta_0$ at ground level (in K). We interpret $\theta_0$ as the background potential temperature, linked exclusively to





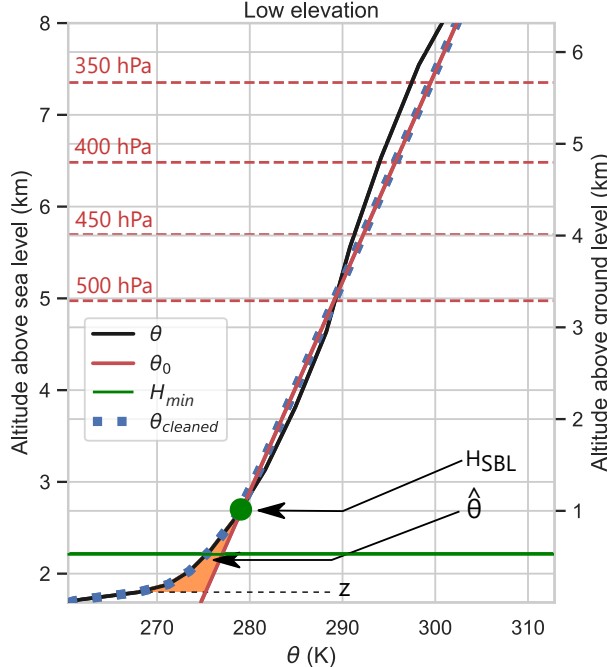

**Figure 2.** Schematic defining variables used for separation of large-scale (background temperature $\theta_0$) and surface components of the vertical potential temperature profile. The solid black line is the typical vertical profile of potential temperature $\theta$ computed for July 2010-2020 at low elevation (120-2300 m) on the transect. Red solid line represents the linear background potential temperature $\theta_0$, which is a linear interpolation of $\theta$ between 350 hPa and the altitude Hmin. The blue dotted line indicates the correction performed on $\theta$ to avoid positive values of the potential temperature deficit $\Delta_\theta = \theta - \theta_0$ above $H_{SBL}$ (green dot), which is the lowest altitude for which $\Delta_\theta$ becomes positive

the large-scale forcing. On the other hand, the difference between the real potential temperature profile $\theta$ and the background temperature $\theta_0$ (called the temperature deficit, $\Delta_\theta = \theta - \theta_0$) is associated with the surface processes (such as katabatic and thermal wind, defined later).

These definitions are based on the hypothesis that we can define a minimum height $H_{min}$, above which the vertical profile of $\theta$ is linear, and the free atmosphere is not influenced by surface processes. The challenge related to the definition of the background potential temperature $\theta_0$ is to be able to accurately define this lowest altitude $H_{min}$ on which to interpolate the potential temperature. Should we take it too low or too high, we would wrongly interpret pressure gradients associated to large-scale processes.

In general, the linear interpolation of $\theta$ between 500 hPa and 350 hPa gives a good first estimate of $\theta_0$, as we are sure that we interpolate $\theta$ above the boundary layer. We take the slope of this linear interpolation, noted $\gamma_{500-350}$, as a first guess of $\gamma_0$.





To be able to interpolate the $\theta$ profile close enough to the surface, we use $\gamma = \frac{\partial \theta}{\partial z}$ the vertical derivative of $\theta$. As a linear vertical profile of $\theta$ means that $\gamma$ should be almost constant, and as $\gamma$ usually becomes much larger that $\gamma_{500-350}$ below a certain height, we define $H_{min}$ as the lowest altitude at which $\gamma$ stays lower than 5 times $\gamma_{500-350}$ (see S3 in the supplement). We also force $H_{min}$ to be greater than 100 m agl, as we assume surface processes to always play a role below this height. Once

135 $H_{min}$ is determined, we calculate $\theta_0$ as the linear interpolation of $\theta$ between $H_{min}$ and 350 hPa, which gives an estimate of $\gamma_0$ and of $\tau_0$ for each 3-hourly time step and each grid cell. Finally, we apply a spatial smoothing function (Gaussian filter) to $\gamma_0$ and $\tau_0$ to obtain a horizontally smooth $\theta_0$, required for the horizontal derivative (see Equation 27 in the Supplementary Materials) in the large-scale wind computation described in Equations (5) and (6). This is a reasonable assumption, since the large-scale potential temperature field does not change abruptly. As $\Delta_\theta$ is the potential temperature deficit in the boundary

140 layer, it must be negative by definition. However, the interpolation line $\theta_0$ always crosses $\theta$. We look for the lowest altitude $H_{SBL}$ (see green dot on Fig. 2) for which delta becomes positive and we force $\Delta_\theta$ to be equal to 0 above this altitude (see blue dotted line on Fig.2. This approximation is justified in section 3.2.

### 2.2.2 Momentum budget decomposition

We use the decomposition of the vertical potential temperature profile to separate the contribution of surface and large-scale

145 pressure gradients in the momentum budget equations. As the wind follows the Antarctic topography at the surface of the ice sheet, we use the momentum budget equations in a coordinate system related to the topography $(x, y, z)$, where $(x, y)$ is the plane following the surface slope of the topography, with $y$ being the downslope direction, and $z$ is the vertical axis normal to the surface slope, as in van den Broeke et al. (2002):

|  | Horizontal advection | Coriolis | Vertical advection & Turbulence | Large-scale | Thermal wind | Katabatic |
|---|---|---|---|---|---|---|
|  | **ADVH** | **COR** | **TURB** | **LSC** | **THWD** | **KAT** |
| Cross-slope: | | | | | | |
| $\dfrac{\partial U}{\partial t} =$ | $-U\dfrac{\partial U}{\partial x} - V\dfrac{\partial U}{\partial y}$ | $+fV$ | $-W\dfrac{\partial U}{\partial z} - \dfrac{\partial \overline{uw}}{\partial z}$ | $-fV_{LSC}$ | $+\dfrac{g}{\theta_0}\dfrac{\partial \hat{\theta}}{\partial x}$ | |
| Downslope: | | | | | | |
| $\dfrac{\partial V}{\partial t} =$ | $-U\dfrac{\partial V}{\partial x} - V\dfrac{\partial V}{\partial y}$ | $-fU$ | $-W\dfrac{\partial V}{\partial z} - \dfrac{\partial \overline{vw}}{\partial z}$ | $+fU_{LSC}$ | $+\dfrac{g}{\theta_0}\dfrac{\partial \hat{\theta}}{\partial y}$ | $+\dfrac{g}{\theta_0}\Delta_\theta \sin(\alpha)$ |

$$(2)$$

155 with

$$\Delta_\theta(z) = \theta(z) - \theta_0(z) \tag{3}$$

$$\hat{\theta}(z) = \int_z^{z_{max}} \Delta_\theta(z)dz \tag{4}$$



The derivative with respect to time of the cross-slope wind $U$ (in $\mathrm{m\,s^{-1}}$) and the downslope wind $V$ (in $\mathrm{m\,s^{-1}}$) are decomposed into six accelerations: the horizontal advection (**ADVH**), the Coriolis deviation (**COR**), the large-scale acceleration (**LSC**), the thermal wind acceleration (**THWD**), the katabatic acceleration (**KAT**) and a residual term that includes the vertical advection, drag and turbulence (**TURB**), in $\mathrm{m\,s^{-1}\,h^{-1}}$. A detailed description of the derivation of these equations is given in the Supplementary Material (Sect. S2.2). The Coriolis factor $f$ is equal to $2\times\Omega\times\sin(\lambda)$ with $\Omega$ the rotation rate of the earth in $\mathrm{s^{-1}}$ and $\lambda$ the latitude. The katabatic acceleration is computed using the potential temperature deficit $\Delta_\theta$ defined in Section 2.2.1 and illustrated in Fig. 2. This is a classic definition documented in Ball (1956) and Mahrt (1982). For the altitude $z$ (agl), if $z > H_{SBL}$, then $\theta = \theta_0$ (as detailed in Section 2.2.1). In the following, we will also use a constant $z_{max}$, an arbitrary height that verifies $z_{max} >> H_{SBL}(x,y,t)$ everywhere, so that we can compute the integration in Equation 4 with constant bounds.

The thermal wind acceleration is a function of the horizontal gradients of $\hat{\theta}$, the vertically integrated potential temperature deficit between the ground and $z_{max}$ (Equation (4) and Fig. 2). It causes a surface flow from areas of weak to large negative values of $\hat{\theta}$, similarly to a sea-breeze circulation, applied to the potential temperature deficit, rather than the potential temperature.

The large-scale acceleration is defined as the geostrophic acceleration in equilibrium with the background potential temperature profile (van den Broeke and van Lipzig, 2003):

$$\frac{\partial U_{LSC}}{\partial \ln(p)} = +\frac{R_d}{f}\left(\frac{p}{p_0}\right)^{\frac{R_d}{C_p}}\left(\frac{\partial \theta_0}{\partial y}\right)_p \tag{5}$$

$$\frac{\partial V_{LSC}}{\partial \ln(p)} = -\frac{R_d}{f}\left(\frac{p}{p_0}\right)^{\frac{R_d}{C_p}}\left(\frac{\partial \theta_0}{\partial x}\right)_p \tag{6}$$

where $p$ is the pressure in Pa, $R_d$ and $C_p$ respectively the gas constant and specific heat capacity of dry air ($R_d$ = 287 $\mathrm{J\,kg^{-1}\,K^{-1}}$ and $C_p$= 1005.7 $\mathrm{J\,kg^{-1}\,K^{-1}}$). The vertical large-scale wind gradient with respect to pressure $\frac{\partial U_{LSC}}{\partial \ln(p)}$ and $\frac{\partial V_{LSC}}{\partial \ln(p)}$ are then integrated between $z$ and $z_{max}$. At $z_{max}$, none of the surface-influenced processes are at stake. Thus, the turbulence, katabatic and thermal-wind accelerations all equal zero. In Antarctica, this happens around 3000 m above ground level

Consequently, at the level $z = z_{max}$, we obtain from Equation (2):

$$U_{LSC}(z_{max}) = \frac{1}{f}\left(\frac{\partial V}{\partial t}(z_{max}) - ADVH_d(z_{max}) + fU(z_{max})\right) \tag{7}$$

$$V_{LSC}(z_{max}) = -\frac{1}{f}\left(\frac{\partial U}{\partial t}(z_{max}) - ADVH_c(z_{max}) - fV(z_{max})\right) \tag{8}$$

$U_{LSC}(z)$ and $V_{LSC}(z)$ are then computed by the integration of Equations 5 and 6, downward from $z_{max}$.



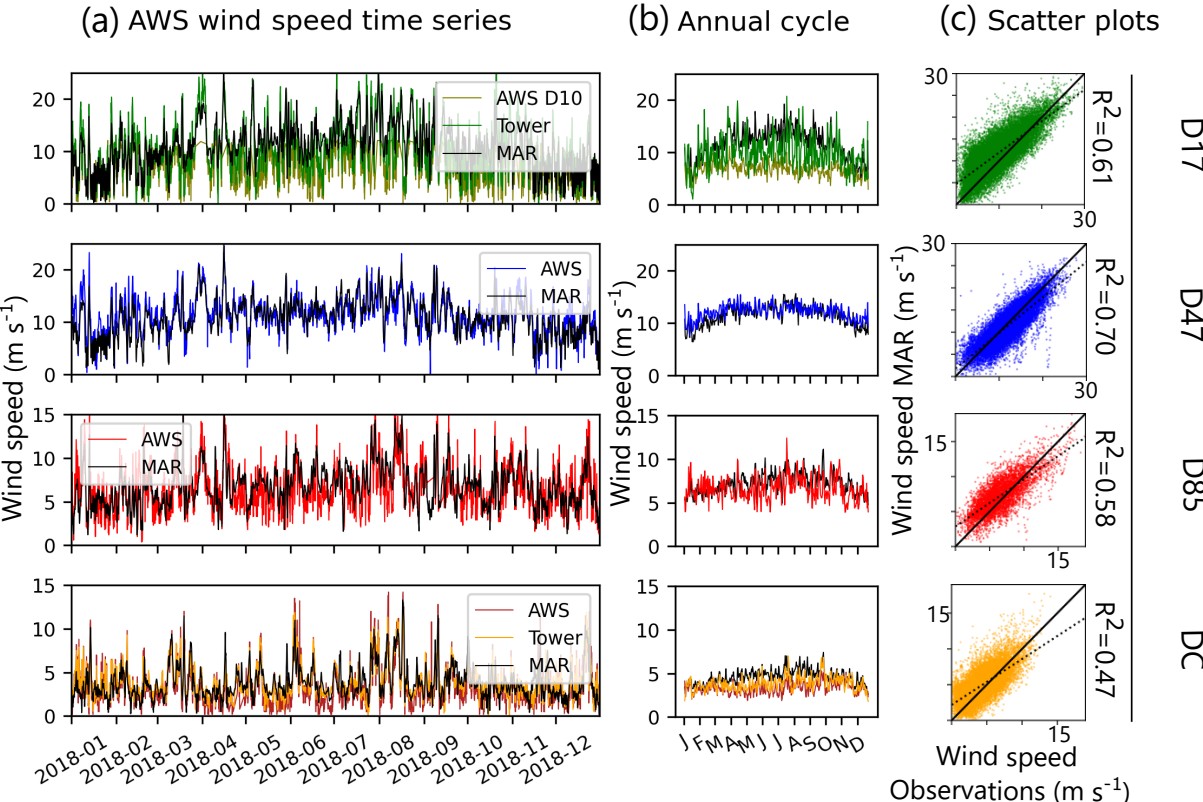

**Figure 3.** From top to bottom D17, D47, D85 and Dome C (a) Comparison of 3-Hourly MAR outputs (black lines) with meteorological tower measurements (when available, i.e. at DC and D17/D10) and AWS (coloured lines). (b) Seasonal cycle computed for the years available in each AWS (see Table 1), with MAR, AWS and the meteorological towers. (c) Scatter plots comparing observations (i.e. meteorological tower for D17 and DC and AWS for D47 and D85) and model outputs for each station. Black solid lines indicate the y=x line while the dotted ones are the linear fit associated with each evaluations. The determination coefficient $R^2$ is indicated next to each scatterplot.

## 3 Evaluation of the model and the method

### 3.1 Evaluation of MAR winds on the transect

Overall, in our simulations, MAR is able to capture the temporal variability of near-surface winds at a 3-hourly frequency reasonably well (Fig. 3a). This includes a good representation of the seasonal cycle (Fig. 3b), which is more pronounced in locations closer to the coast, such as D17 and D47. The model underestimates slightly the mean 2-m wind speed at D47 with a bias of -0.6 m s$^{-1}$. However, across all the other stations, the model tends to overestimate the mean wind-speed with a bias ranging from 0.6 m s$^{-1}$ for D85 to 2.00 m s$^{-1}$ at D17. The strongest correlations are found in locations with higher average wind speeds such as D47 ($R^2 = 0.7$) and D17 ($R^2 = 0.61$) (Fig. 3c). Turbulence above the surface boundary layer



in MAR is parameterised using a local E–$\epsilon$ scheme, adapted for stable atmospheric boundary layers in which small eddies develop and dissipate rapidly. Local turbulence schemes, however, commonly fail to represent the downward entrainment of
momentum by large eddies of greater vertical extent (Hillebrandt and Kupka, 2009). This typically happens in well-mixed atmospheric boundary layers, as encountered in coastal Adélie Land during strong winds (Amory et al., 2017). The resulting misrepresentation of wind speed maxima is partly compensated by a temperature-dependent parameterization for $z_0$, which has been tuned to better capture observed wind speed maxima (at the expense of minima) and seasonal variations in wind speed in coastal Adélie Land (Amory et al., 2021).

## 3.2 Evaluation of the momentum budget decomposition (MBD)

The momentum budget decomposition (MBD) performs a separation between the accelerations of the wind induced by large-scale forcings (**LSC**) that are the only drivers above the boundary layer and the accelerations of the wind resulting from surface forcings (i.e. katabatic (**KAT**), thermal wind (**THWD**) and turbulence (**TURB**)), that are zero above the boundary layer and are intensified near the surface. **LSC** is computed using the background potential temperature $\theta_0$, while surface processes are
computed using the potential temperature deficit $\Delta_\theta$ for **KAT** and the integrated potential temperature deficit $\hat{\theta}$ for **THWD**. As a first evaluation step, we verify that this is indeed the case by plotting vertical profiles of each acceleration of the wind (Fig. S4) and of the different metrics ($\theta$, $\theta_0$, $\Delta_\theta$, $\hat{\theta}$) computed during our separation of the vertical potential temperature on the transect (Fig. 4): the katabatic acceleration is proportional to $\Delta_\theta$ which is intensified near the surface and decreases exponentially with height; the turbulence has a local maximum slightly above the surface; and at higher elevation, the large-
scale forcing is balanced by the Coriolis acceleration, all other terms being near zero. The vertical profiles are qualitatively similar to those in van den Broeke et al. (2002), who performed the same decomposition in the Droning Maud Land sector of Antarctica.

In addition, we find the total pressure gradient force (PGF) to be well reproduced by our decomposition. The total pressure gradient force is the sum of katabatic, large-scale and thermal wind accelerations (Sect. S2.1).

$$\mathbf{PGF} = \mathbf{LSC} + \mathbf{KAT} + \mathbf{THWD} \tag{9}$$

We compute the PGF (Equation 9), and compare it to the PGF natively computed by the MAR model, at each 3-hourly timestep. Fig. 5 shows this comparison at D47, the site with the largest katabatic acceleration for August 2012. This month was chosen because it displays two consecutive high wind-speed events that are detailed in section 3.3. The other stations are shown in Fig. S6.

The MBD captures well the temporal variations and extrema of the pressure gradient force (Fig. 5). Some of the maxima are underestimated (at D47, our MBD exhibits a mean bias of -1.3 $\mathrm{m\,s^{-1}\,h^{-1}}$). This is due to the fact that the background potential temperature profile ($\theta_0$) is approximated by a linear slope, which is not always exactly the case, and causes an under-estimation of the large-scale acceleration, particularly near the coast (D17), where the vertical structure of air masses is more complex. Quantitatively, the normalized root mean square error (NRMSE, i.e. the root mean square error between MAR PGF and our
MBD PGF, normalized by the maximum value minus the minimum value of the time series of MAR PGF at each grid-cell)





**Figure 4.** Vertical profiles on the transect averaged for the months of July 2010-2020 of (a) potential temperature ($\theta$), (b) background potential temperature ($\theta_0$), (c) potential temperature deficit ($\Delta_\theta$) (d) vertically integrated potential temperature deficit ($\hat{\theta}$), (e) norm of wind speed (|WS|), (f) norm of large-scale acceleration (|LSC|), (g) norm of katabatic acceleration (|KAT|), (h) norm of thermal wind acceleration (|THWD|).



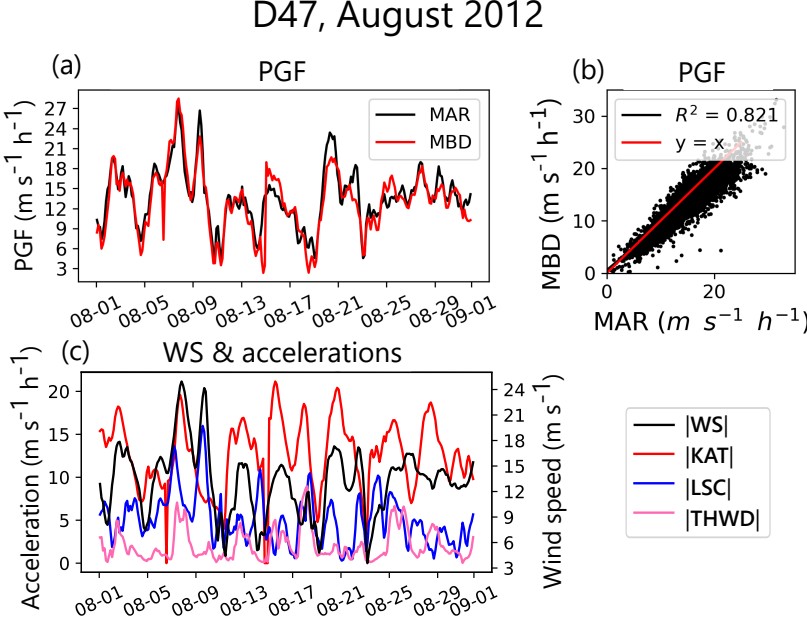

**Figure 5.** Comparison of MAR Pressure Gradient Force (PGF) output with our MBD PGF at D47 at the surface. (a) 3-hourly time serie comparison of MAR PGF versus MBD PGF for a winter month (August 2012). (b) Scatter plot of 3-hourly MAR PGF versus MBD PGF for the winter months (June, July, August) 2010-2020. (c) Wind speed (solid black line) and accelerations used to compute the PGF (katabatic acceleration in red, large-scale acceleration in blue and thermal-wind acceleration in pink).

was about 7.5 % for July 2010-2020 at the surface. The coefficient of determination between the July datasets is relatively high everywhere on the transect ($R^2 > 0.6$), with values ranging from 0.61 at D17 to 0.93 at DC. It indicates a good correlation between our MBD and MAR outputs, and shows that the MBD is internally consistent. The approximations described in section 2.2.1 do not introduce significant errors.

### 230   3.3   Evaluation of the Momentum Budget Decomposition (MBD) during a high wind speed event

MAR MBD is performed in August 2012, for two successive high wind speed events (HWSE). HWSE are defined as days for which the total wind speed is greater than the $90^{th}$ percentile of the 10 years 3-hourly dataset. During the first event on August $7^{th}$, the katabatic layer (the air mass cooled down by the surface) starts growing around 00:30 (UTC), reaches its maximum around 19:30 (UTC) (Fig. 6a) and decreases during the next 24 hours. This is accompanied by an increase of

background temperature (up to 288 K, Fig. 6b), which, combined with the low potential temperature, creates a strong potential temperature inversion ($\Delta_\theta$ = -22.2 K, Fig. 6c) and vertically integrated potential temperature deficit(Fig. 6d). This katabatic layer development is characteristic of a katabatic event (Vihma et al., 2011). It is consistent with the computed katabatic acceleration (Fig. 6g), which develops and reaches a maximum on that day, while the large-scale acceleration does not exhibit





**Figure 6.** Time series of vertical profiles during two high wind speed events at D47 on August $7^{th}$ and $9^{th}$, 2012 (denoted by vertical dotted lines) (a) potential temperature ($\theta$), (b) background potential temperature ($\theta_0$), (c) potential temperature deficit ($\Delta_\theta$), (d) vertically integrated potential temperature deficit ($\hat{\theta}$), (e) norm of total wind speed (|WS|), (f) norm of large-scale acceleration (|LSC|), (g) norm of katabatic acceleration (|KAT|), (h) norm of thermal wind acceleration (|THWD|)



any significant increase. As a conclusion, the strong wind speed maximum on August $7^{th}$ is primarily driven by the katabatic
acceleration, and we consider it to be a katabatic-driven event.

On the other hand, two days later, on August $9^{th}$, another peak of wind speed extends much higher in the atmosphere.
This time, the temperature deficit at the surface is limited ($\Delta_\theta$ = -8.0 °, Fig. 6c), and the katabatic acceleration, while present,
remains limited. The large-scale acceleration, however, increases progressively, starting on August $8^{th}$, from 16:30 UTC to its
maximum on August $9^{th}$ at 13:30 UTC (Fig. 6f), just before the wind speed maximum around 19:30 UTC (Fig. 6e). Therefore,
this high wind speed event is attributed mainly to large-scale forcing.

As a conclusion, our MBD produces logical results in regards to the vertical structure of the atmosphere. It also confirms
hints of katabatic events, visible in the development of the katabatic layer in the vertical profile of potential temperature, and
provides us with additional information regarding synoptic events, enabling us to clearly identify the main driver of these
HWSE. It also underlines the necessity of studying these events at a 3-hourly time-scale in order to be able to capture the
variations of the katabatic layer and the large-scale acceleration.

## 4   Results

### 4.1   Quasi-stationary momentum budget and dominant components

The seven terms in the momentum budget equations (5) and (6) do not share equal roles in shaping the wind speed intensity nor
variability. Three of them, katabatic, thermal wind and large-scale, can be viewed as active terms because they are produced
by a forcing, either large-scale or surface pressure-gradients. By opposition, turbulence, Coriolis and advection accelerations
can be viewed as passive terms as they only come into play when the motion has been triggered by an active term.

We evaluated the dominant terms in the surface momentum budget by looking at the average amplitude of each accelera-
tion, computed on 3-hourly outputs for the period 2010-2020, in summer (December-January-February, DJF), winter (June-
July-August, JJA), and annual mean, shown on Table 3. The temporal derivative of the wind vector, $|\nabla_t \mathbf{WS}|$, is 5 orders of
magnitude smaller than the other accelerations. Therefore we can assume a quasi-stationary momentum budget everywhere on
the transect, and the total wind speed $|\mathbf{WS}|$ is directly related to the norm of the sum of the other accelerations through the
quasi-geostrophic equilibrium:

$$\mathbf{COR} + \mathbf{LSC} + \mathbf{THWD} + \mathbf{KAT} + \mathbf{ADVH} + \mathbf{TURB} \approx 0$$

$$\Rightarrow \mathbf{WS} = \mathbf{V_{LSC}} + \mathbf{V_{THWD}} + \mathbf{V_{KAT}} + \mathbf{V_{ADVH}} + \mathbf{V_{TURB}} \tag{10}$$

$$\Leftrightarrow |\mathbf{WS}| = \frac{1}{f}|\mathbf{LSC} + \mathbf{THWD} + \mathbf{KAT} + \mathbf{ADVH} + \mathbf{TURB}| \tag{11}$$

with $\mathbf{V_{MBD}} = -\boldsymbol{f}/f^2 \times \mathbf{MBD}$ being the geostrophic wind equivalent to each $\mathbf{MBD}$ acceleration, i.e. the stationary wind
vector that would result from a balance of the acceleration under consideration with the Coriolis acceleration.

The katabatic, large-scale and turbulent accelerations are the three dominant terms (Table 3). These three terms alone in
Equation 10 are enough to reproduce the direction and intensity of the near-surface wind (Fig. S5). Horizontal advection and





| Name | \|**KAT**\| | | | \|**LSC**\| | | | \|**THWD**\| | | | \|**ADVH**\| | | |
|------|-----|-----|-----|-----|-----|-----|-----|-----|-----|-----|-----|-----|
| | DJF | JJA | Ann | DJF | JJA | Ann | DJF | JJA | Ann | DJF | JJA | Ann |
| D17 | 8.6 | 18.9 | 12.7 | 4.3 | 5.4 | 4.8 | 1.3 | 4.6 | 2.6 | 2.4 | 5.5 | 4.2 |
| D47 | 7.3 | 12.3 | 9.3 | 3.7 | 4.5 | 4.1 | 0.6 | 1.3 | 0.9 | 0.9 | 1.6 | 1.3 |
| D85 | 4.3 | 6.5 | 5.13 | 3.7 | 5.5 | 4.6 | 1.5 | 2.3 | 1.7 | 0.4 | 0.7 | 0.6 |
| DC | 0.3 | 0.5 | 0.4 | 3.1 | 4.3 | 3.7 | 0.5 | 0.7 | 0.6 | 0.3 | 0.6 | 0.5 |

| Name | \|**COR**\| | | | \|**TURB**\| | | | $\|\nabla_t\mathbf{WS}\|$ ($\times 10^{-5}$) | | |
|------|-----|-----|-----|-----|-----|-----|-----|-----|-----|
| | DJF | JJA | Ann | DJF | JJA | Ann | DJF | JJA | Ann |
| D17 | 4.0 | 6.7 | 5.6 | 8.0 | 19.3 | 14.4 | 13.9 | 11.1 | 11.1 |
| D47 | 4.5 | 6.3 | 5.6 | 5.3 | 10.4 | 8.2 | 1.0 | 5.6 | 5.6 |
| D85 | 3.0 | 4.0 | 3.6 | 3.3 | 6.1 | 4.9 | 8.3 | 5.6 | 5.6 |
| DC | 1.9 | 2.6 | 2.3 | 1.8 | 2.9 | 2.5 | 5.6 | 5.6 | 5.6 |

**Table 3.** Averaged 2010-2020 summer (DJF), winter (JJA) and annual (Ann) norm of accelerations: katabatic (**KAT**), large-scale (**LSC**), thermal wind (**THWD**), horizontal advection (**ADVH**), Coriolis (**COR**), turbulent accelerations (**TURB**) and derivative with respect to time of the wind speed ($\|\nabla_t\mathbf{WS}\|$), on the 4 stations of the transect. The seasonal values are computed in $\mathrm{m\,s^{-1}\,h^{-1}}$. Norms are computed with MAR 3-hourly outputs.

thermal wind accelerations have lower magnitudes, but become significant with regard to the other terms close to the coast (D47 and D17) and on the ocean (Fig. 7). In the rest of the study, special attention will be given to the dominant terms of the momentum budget: katabatic, large-scale, thermal wind and turbulence.

## 4.2 Drivers of spatial wind variability

In Antarctica, the wind speed generally increases from the plateau to the coast (Fig. 1). On the transect, mean July 2010-2020
3-hourly MAR wind speed are ranging from 4.9 to 14.1 $\mathrm{m\,s^{-1}}$, with a spatial standard deviation of 2.6 $\mathrm{m\,s^{-1}}$. During summer, mean wind speeds are lower, ranging from 3.6 to 9.1 $\mathrm{m\,s^{-1}}$ with a spatial standard variation reduced to 1.7 $\mathrm{m\,s^{-1}}$.

The katabatic acceleration is proportional to the surface slope and to the potential temperature deficit (Equation 2). On the plateau, although the potential temperature deficit $\Delta_\theta$ is large (Fig. 7f), the slope is near zero, and the katabatic acceleration is negligible (Fig. 7a). The katabatic acceleration increases strongly in a band of 250 km along the coast, where the surface
slope is significant. We refer to this narrow band of strong katabatic acceleration as the active katabatic belt. Here, we want to emphasise that the katabatic acceleration points in the slope direction. Consequently, in the quasi-geostrophic stationary conditions detailed in Section 4.1 it increases the wind speed in the cross-slope direction, along the elevation contours (Fig. 7a).

There is a secondary, narrower active thermal wind belt starting ~100 km from the coast (Fig. 7c), in which the thermal wind
opposes the katabatic acceleration most of the time. This is a consequence of the pressure low created by the displacement of



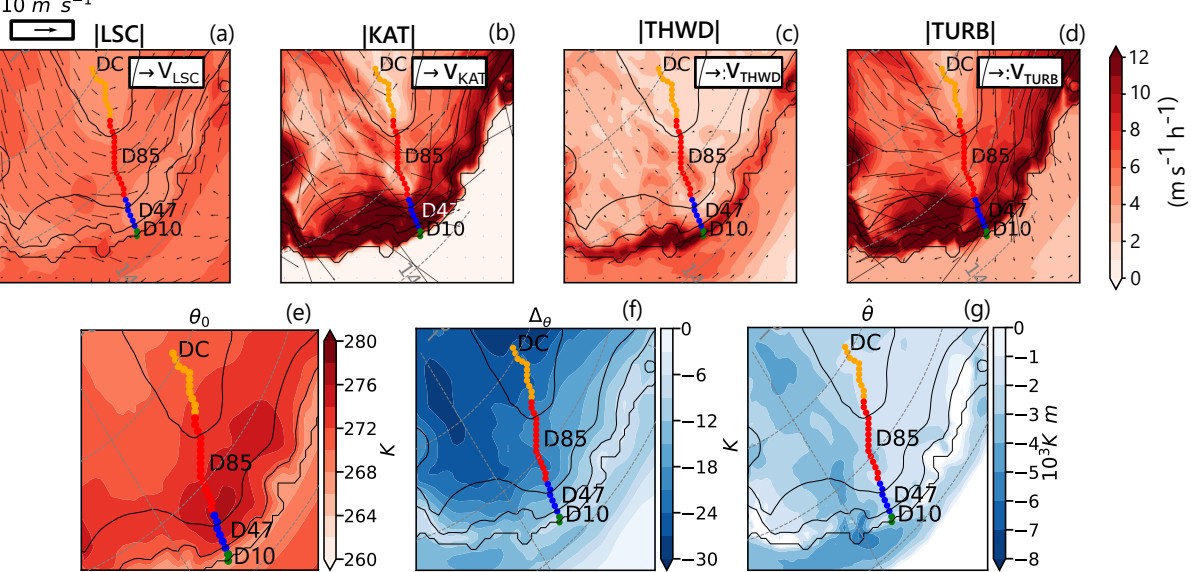

**Figure 7.** (Upper-panel) Mean July 2010-2020 norm of accelerations at surface level (∼7 m agl) computed with 3-hourly MAR outputs:(a) large-scale, (b) katabatic, (c) thermal wind and (d) turbulence. Superimposed are the equivalent wind vectors. (Lower panel) Mean July 2010-2020 values of the background temperature $\theta_0$ (e), the potential temperature deficit $\Delta_\theta$ (f) and the vertically integrated potential temperature deficit $\hat{\theta}$ (g) at surface level (∼7 m agl) computed with 3-hourly MAR outputs.

cold air from the inland to the coast. It implies a secondary circulation (thermal wind) in the opposite direction (Parish et al., 1993).

The large-scale acceleration (Fig. 7b) is spatially more uniform than the katabatic acceleration (Fig. 7a). The large-scale polar circulation cell is characterised by a high surface pressure on the plateau and lower pressure on the coast. In addition, we

find that, on average, the large-scale surface pressure-gradient is aligned with the topography but unlike the katabatic forcing, its value is not directly proportional to the slope angle. The mean magnitude of the large-scale acceleration is weaker than the katabatic term everywhere on the transect, except at Dome C (Table 3). This weaker mean intensity is due to the changing location of synoptic perturbations. In winter, at D47, for instance, the large-scale acceleration displays a mean value of 5.4 $\mathrm{m\,s^{-1}\,h^{-1}}$, but a value of the 99th percentile (computed with 3-hourly outputs) of about 12.6 $\mathrm{m\,s^{-1}\,h^{-1}}$, which is comparable

to the mean value of the katabatic acceleration for that period.

The turbulent acceleration mostly encompasses drag and vertical advection (supposed negligible by van den Broeke and van Lipzig (2003)). The drag is proportional and in the opposing direction to the wind vector (Fig. 7d).

To sum up, the mean acceleration of the wind on the slope of the plateau is dominated by the katabatic forcing, but the large-scale forcing also plays a role, as it has the same spatial pattern, and the same sign, albeit with a smaller amplitude in the

active katabatic belt. These two forcings are opposed by turbulence, and by thermal wind very close to the coast, causing the wind speed maximum to be slightly more upslope than the slope would dictate alone.



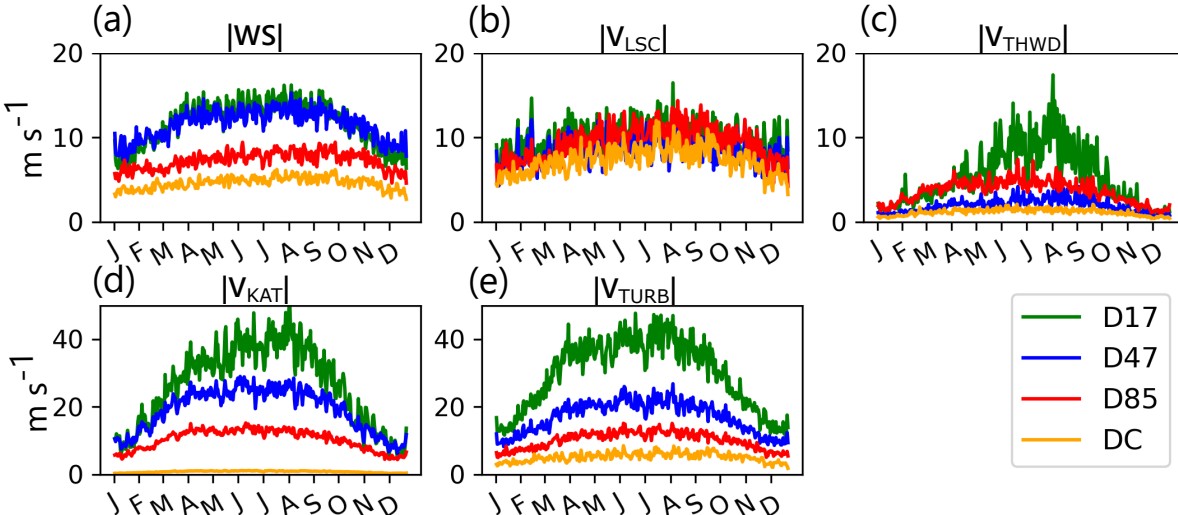

**Figure 8.** Seasonal cycle of 3-hourly winds averaged over 10 years for total wind speed (a), wind speed equivalent to large-scale acceleration (b), wind speed equivalent to thermal wind (c), wind speed equivalent to katabatic (d) and wind speed equivalent to turbulent accelerations (e). Note that the y-axis is different between the top panel (|WS|, $|V_{LSC}|$, $|V_{THWD}|$) and the bottom panel ($|V_{KAT}|$, $|V_{TURB}|$).

### 4.3 Drivers of seasonal wind variability on the transect

The wind speed displays a seasonal cycle that peaks in late winter (August to September) and is especially pronounced in the low elevation and coastal areas. In Fig. 8, we compute the annual cycle of the total wind speed (average of 3-hourly time steps for 2010-2020), and of wind speed equivalent to large-scale acceleration ($\mathbf{V_{LSC}}$), thermal wind ($\mathbf{V_{THWD}}$), katabatic ($\mathbf{V_{KAT}}$) and turbulent accelerations ($\mathbf{V_{TURB}}$). Below 1500 m asl, the seasonal amplitude in wind speed between summer and winter ($\Delta|\mathbf{WS}|_{\mathbf{JJA-DJF}}$ equals 5.6 m s$^{-1}$ at D17 and 3.8 m s$^{-1}$ at D47) is larger than the July standard deviation of 3-hourly July wind speed (highest variability during winter months) computed over the 10 years dataset: ($\sigma_{|\mathbf{WS}|}$ equals 4.1 m s$^{-1}$ at D17 and 3.4 m s$^{-1}$ at D47). In higher elevation and interior zones, the seasonal cycle is much weaker and the 10-years standard deviation of July 3-hourly wind speed exceeds $\Delta|\mathbf{WS}|_{\mathbf{JJA-DJF}}$.

Because of the strong seasonal cycle of the temperature deficit, as expected, a similar behaviour for katabatic and thermal winds (which are directly related to the surface inversion) is found. Katabatic winds have a strong seasonal cycle (Fig. 8d) which peaks in August and is increasingly stronger from the inland to the coast. The strongest seasonal amplitude is found at D17 ($\Delta|\mathbf{V_{KAT}}|_{\mathbf{JJA-DJF}}$ is 25 m s$^{-1}$). Note that the seasonal amplitude of katabatic winds is significantly stronger than that of the total wind speed, because it is damped by turbulence, which also displays a strong seasonal cycle ($\Delta|\mathbf{V_{TURB}}|_{\mathbf{JJA-DJF}}$



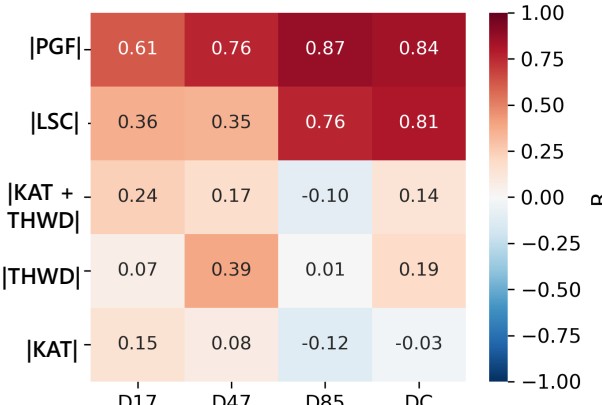

**Figure 9.** Correlation coefficient (R) between the 3-hourly total wind speed and the the different accelerations in July 2010-2020.

is 22 m s$^{-1}$). Thermal wind also depends on the inversion layer but is concentrated near the coastline (Fig. 7c), and shows a strong seasonality for D17 exclusively ($\Delta|\mathbf{V_{THWD}}|_{\mathbf{JJA-DJF}}$ is 3.6 m s$^{-1}$).

Surprisingly, the thermal wind is stronger at D85 than at D47, closer to the coast. This is due to the small valley shape
around D85 (Fig. 1b) that enables piling up of cold air coming from the plateau (Fig. 4d), while D47 is located in the middle of a steep slope. Unlike surface-related momentum contributions, large-scale winds exhibit a weak seasonal cycle, identical for all stations, with $\Delta|\mathbf{V_{LSC}}|_{\mathbf{JJA-DJF}}$ ranging from 1.4 for D47 to 2.7 m s$^{-1}$ for D85. Therefore the large-scale contribution is unlikely to explain the seasonal variability of the total wind speed, nor the spacial differences in the seasonal cycle along the transect.

From these analyses, and from supplementary spectral analyses (Fig. S9), we conclude that the seasonal variability of wind speed is mainly produced by the seasonal cycle of katabatic acceleration, which is proportional to the surface inversion strength. The large-scale forcing only plays a minor role in the seasonal cycle of near-surface wind.

## 4.4 Drivers of 3-hourly winter variability

In this section, we investigate the drivers of near-surface wind variability at the synoptic scale. We analyse high temporal
resolution wind speed outputs (3-hourly) for the months of July 2010-2020, when wind speeds are particularly high (seasonal maximum) and the diurnal cycle is very weak (polar night). We use the correlation coefficients between the different accelerations and the total wind speed to identify the dominant drivers of wind speed variability (Fig. 9).

In the regions where the katabatic acceleration is small (D85 and DC, see Table 4 or Fig. 9), as expected, the correlation coefficient between the large-scale acceleration and the total wind speed is very high (R>0.75). Closer to the coast, this correlation
coefficient decreases, reaching ∼0.35 for both D17 and D47. There, although the katabatic becomes stronger (greater than 12 m s$^{-1}$ h$^{-1}$ on average in the winter, more than twice the value of the mean large-scale acceleration, Table 4), the acceleration





| Name | $\|\mathbf{KAT}\|$ | | | $\|\mathbf{LSC}\|$ | | | $\|\mathbf{THWD}\|$ | | | $\|\mathbf{KAT+THWD}\|$ | | | $\|\mathbf{PGF}\|$ | | |
|------|------|----------|------|------|----------|------|------|----------|------|------|----------|------|------|----------|------|
| | Avg. | $\sigma$ | R | Avg. | $\sigma$ | R | Avg. | $\sigma$ | R | Avg. | $\sigma$ | R | Avg. | $\sigma$ | R |
| D17 | 18.9 | 8.1 | 0.15 | 5.4 | 3.3 | 0.36 | 4.3 | 4.2 | 0.07 | 16.5 | 6.2 | 0.24 | 16.4 | 6.1 | 0.61 |
| D47 | 12.2 | 3.6 | 0.08 | 4.6 | 2.7 | 0.35 | 1.2 | 1.0 | 0.39 | 12.5 | 3.7 | 0.17 | 13.1 | 4.0 | 0.76 |
| D85 | 6.04 | 1.7 | -0.12 | 5.5 | 2.8 | 0.76 | 2.3 | 1.3 | 0.01 | 4.7 | 1.3 | -0.1 | 8.4. | 2.9 | 0.87 |
| DC | 0.5 | 0.2 | -0.03 | 4.4. | 2.3 | 0.84 | 0.7 | 0.6 | 0.19 | 4.7. | 0.6 | 0.14 | 4.5 | 2.3 | 0.84 |

**Table 4.** July 2010-2020 statistics for katabatic (KAT), large-scale (LSC), thermal wind (THWD), surface processes (KAT + THWD) and total pressure gradient force (PGF) accelerations, on the 4 stations on the transect. The averaged value (Avg.) and standard deviation ($\sigma$) are computed in $\mathrm{m\,s^{-1}\,h^{-1}}$. R is the correlation coefficient with the total wind speed. All metrics are computed with MAR 3-hourly outputs for July 2010-2020.

remains poorly correlated with the total wind speed (R respectively equals 0.15 and 0.08 for D17 and D47). At these specific locations, it seems that none of the decomposed accelerations singularly dominate the 3-hourly wind speed variability.

Before explaining these results on the transect, we want to test how representative of the coastal region of Adélie land our

transect is. Therefore, we analyse the correlation coefficient between the katabatic acceleration and the total wind speed, not only on the transect, but rather on a surrounding region of 1800 km×1550 km centered on Adélie Land (Fig. 10a). In the active katabatic belt, some regions show a higher correlation (R>0.5) between the katabatic acceleration and the total wind speed. Our transect is located right in the middle between two of these regions, meaning that it is not necessarily representative of the whole region.

A first explanation to the low correlation between the katabatic acceleration and the total wind speed could be that, close to the coast, the thermal wind opposes the katabatic acceleration. Thus, the sum of katabatic and thermal wind accelerations ($\|\mathbf{KAT+THWD}\|$, in other words, surface processes) displays a better correlation with the total wind speed (R=0.24 at D17 and R=0.17 at D47) than the katabatic acceleration alone ($\|\mathbf{KAT}\|$, R=0.15 at D17 and R=0.08 at D47; Table 4, Fig. 9). Considering surface processes $\|\mathbf{KAT+THWD}\|$ together prevents us from overestimating the impact of the katabatic

acceleration, especially in cases where both thermal wind and katabatic acceleration are large but of opposite direction.

In order to test this hypothesis on the whole region, we compute the scalar product of mean July 2010-2020 thermal-wind with the wind direction. It enables us to assess whether thermal wind actively opposes the wind (negative values) or act in a direction that increases it (positive values, Fig. 10f).

We observe that out of the four zones of higher correlation between the katabatic acceleration and the total wind speed

(R>0.5, Fig. 10a), three of them correspond to locations where the action of thermal wind is positive (II, III, IV). In the last one (I), the scalar product of total wind speed and thermal wind is close to zero, indicating that the thermal wind has no effect on the total wind speed. Note that some area of positive action of thermal wind and strong katabatic accelerations (e.g. west and south of (I)) don't create the conditions for a strong correlation of the katabatic acceleration and the total wind speed.





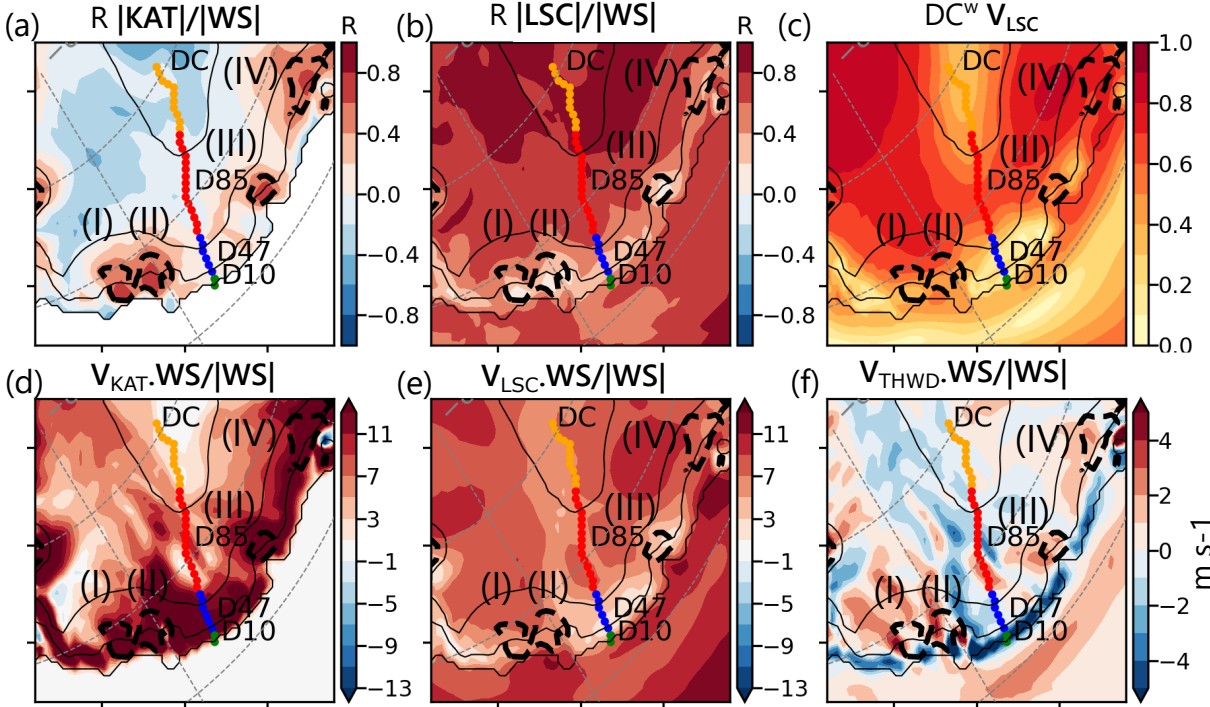

**Figure 10.** (a) Average July 2010-2020 correlation coefficient of 3-hourly katabatic acceleration and wind speed (b) Average July 2010-2020 correlation coefficient of 3-hourly large-scale acceleration and wind speed (c) directional constancy of 3-hourly large-scale wind speed. (d, e, f): Mean of 3-hourly July 2010-2020 scalar product normalised by the norm of wind speed of (d) 3-hourly katabatic wind speed and total wind speed, (e) 3-hourly large-scale and total wind speed, (f) 3-hourly thermal-wind and total wind speed. For the 6 panels, the dotted black line corresponds to the line for which the correlation coefficient of katabatic acceleration and total wind speed reaches 0.5. Four zones of higher correlations are indicated: (I), (II), (III) and (IV)

Therefore, the acceleration provided by thermal wind is not sufficient to fully explain the locations of the highest values of the correlation between the katabatic acceleration and the total wind speed.

While the katabatic acceleration is always directed downslope, the large-scale wind speed displays a much more variable direction, indicated by low values of directional constancy ($DC^w V_{LSC}$) (see Fig. 10 c). $DC^w$ is computed as follows:

$$DC^w = \frac{\sqrt{\overline{U}^2 + \overline{V}^2}}{\frac{1}{N}\sum_{i=1}^{N}|WS_i|} \qquad (12)$$

On the plateau, $DC^w$ is close to zero, which is typical of a wind with no preferred direction. In the valleys around and in the higher elevation zone, winds tend to blow in a preferred downslope direction and $DC^w$ is closer to 1. However, from D47 to the coast, $DC^w$ falls back to zero. That part of the segment is located on a ridge. As a result, the topographic steering of surface pressure gradient is less important than in the valleys. In locations with small $DC^w$ (i.e. on ridges and plateaus), the




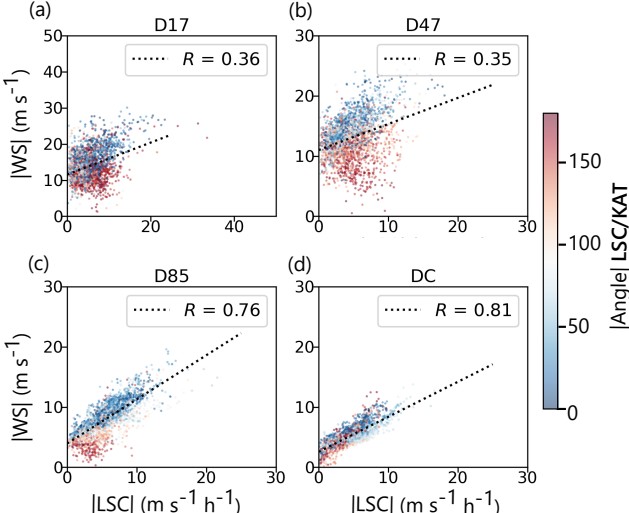

**Figure 11.** Correlations for July 2010-2020, 3-hourly, between the large-scale acceleration and the total wind speed at (a) D17 (coast), (b) D47, (c) D85, (d)DC (plateau). The colorbar indicates the angle between the katabatic and the large-scale acceleration Around 0 °, LSC and KAT are aligned, around 180 °, they are of opposite direction.

large-scale pressure gradient sometimes opposes the katabatic acceleration, leading to decreased correlations of the katabatic acceleration with total wind speed.

The angle between the large-scale acceleration and the topography seems to have a major impact on the wind speed intensity. This is confirmed by Fig. 11, where we find a clear partition of the influence of the angle: the wind speed is higher when the large-scale acceleration is aligned with the topography (angle 0°, blue on Fig. 11), and weaker when the large-scale opposes the katabatic acceleration (angle 180°, red dots on Fig. 11). In locations where the large-scale direction is highly variable (e.g. between D10 and D47, close to the coast), the angle between katabatic and large-scale displays both positive and negative

values. In this situation, there is not a single driver of the wind speed intensity, but rather a competition between katabatic acceleration (mainly in winter and at night), and large-scale forcing which is particularly effective when it is aligned with the katabatic acceleration. Therefore, it is essential to compute the momentum budget decomposition in order to identify the drivers of wind speed variability.

To sum up, the dominant drivers of synoptic scale variability depend on the location. On the Plateau, the large-scale forcing

logically dominates the variability. In the active katabatic belt, the katabatic acceleration has the strongest amplitude and variance (Table 4). However, a strong katabatic forcing is not always causing high wind speed, because the large-scale acceleration can counteract the katabatic acceleration, if it is oriented upslope. Thus, the angle between the large-scale acceleration and the surface slope is a key factor in explaining strong wind speed events in coastal Antarctica: the highest wind speed events happen when the katabatic and large-scale forcing are aligned (Fig. 11), although each acceleration, when acting alone, can also cause

strong wind speed (Fig. 6). On the coast, the pile-up of cold air at sea level counteracts the katabatic forcing, which explains





why the strongest wind speeds are not found right on the coast (Fig. 7). There, all of the terms of the momentum budget are important, and there isn't a dominant forcing term. We demonstrated that, although the katabatic term is the dominant contributor to the mean wind speed, spatially, and seasonally; at the event scale, accelerations are more complex, and wind events cannot systematically be interpreted as katabatic.

## 5   Discussion

In this study, we obtained a comprehensive understanding of the drivers of East Antarctic near-surface winds by combining directional consistency and momentum budget decomposition analyses. As different accelerations can cancel each other when in opposed directions, the consistent directional behavior of the wind serves as a valuable complementary tool to the MBD for examining the drivers of near-surface winds in the active katabatic region. It reveals locations where correlation of the katabatic acceleration with the wind speed is weak due to variable large-scale wind direction. However, we show that relying solely on directional constancy does not provide a reliable diagnosis of near-surface wind drivers, because large-scale winds exhibit areas of significant directional consistency in regions where katabatic acceleration is low and does not correlate with wind speed, in line with Parish and Cassano (2003).

Previous work using the MBD had focused on monthly averages ((van den Broeke and van Lipzig, 2003), (Bintanja et al., 2014)). However, to understand the drivers of high wind speed events, it is necessary to study winds at a sub-daily resolution. Here we have demonstrated that variations in the temperature deficit strength or in large-scale pressure gradient occur within a day (e.g. Fig. 6). Therefore, to highlight the influence of synoptic events on the nature of near-surface winds in the active katabatic belt, we have selected a 3-hourly time-step. In this pursuit, we have adapted the method for extrapolating the free-atmosphere vertical potential temperature profile $\theta_0$ developed by van den Broeke and van Lipzig (2003) for monthly outputs. However, the linearization of the vertical potential temperature profile is challenging with 3-hourly outputs. Most of the profiles featuring a large Normalized Root Mean Square Error (NMRSE) between the native MAR PGF and our MBD PGF, i.e. greater than the 90% percentile (which corresponds approximatively to a NRMSE greater than 10 %, Fig. S7) do not feature any abrupt increase in the vertical derivative of potential temperature at the top of the inversion layer, leading to an underestimation of the MBD PGF. Some other profiles display intrusions of air-masses (characterized by a non strictly monotonous profile of potential temperature) or a secondary linear section with a different slope under 500 hPa. Examples of these types of profiles are shown in Figure S8. From the comparison between $PGF_{MAR}$ and our $PGF_{MBD}$, our MBD works better in the interior than close to the coast, where these types of profiles are more likely to be found, probably due to the vicinity of the ocean. Overall, on our transect, there is a satisfactory low number of profiles exhibiting large NRMSE. Increasing the temporal resolution of our dataset would be even more challenging. The vertical profiles of potential temperature would be even less smooth and hard to interpolate. Furthermore, the stationary approximation has been made at a 3-hourly time-scale, which is generally valid (Section 4.1), but would not be accurate at a finer resolution.

Finally, it is crucial to have a good depiction of the vertical structure of the atmosphere, inside and above boundary layer, to study the drivers of near-surface winds. Our regional atmospheric model has been evaluated at 2 m agl and performs well



at that height. However, its ability to accurately represent vertical atmospheric profiles has not been assessed due to limited observations, only available at DC where there is no katabatic acceleration, and DDU where the performance of MAR is limited. In the future, it would be valuable to have available radiosoundings in a katabatic-active region to conduct an observational study about the drivers of near-surface winds and to evaluate more accurately our model.

## 6 Conclusions

To understand the drivers of near-surface winds in Antarcica, we have separated the contributions to wind speed of surface-based and large-scale pressure gradients, using the momentum budget decomposition. We focused on a well instrumented transect running through Adélie Land (east Antarctica), from the plateau to the coast. We demonstrated that seasonal and spatial variability of near-surface winds in Adélie Land are both dominated by surface processes, notably by katabatic winds. At a 3-hourly time-scale however, on our study transect, identifying the main driver of the wind becomes more challenging. Large-scale pressure acceleration correlates well to the total wind speed from the plateau to ∼250 km from the coast, in locations where the katabatic acceleration is weak to null. Then, in the active katabatic and thermal wind belt, below 2000 m asl, surface processes come into play and decrease the correlation of large-scale acceleration with the total wind speed. Due to the highly varying angle between large-scale and katabatic accelerations, close to the coast, the two are often in competition. Thus, correlation coefficients of large-scale and katabatic processes with total wind speed remain low, weaker than respectively 0.4 and 0.2. In that region of the transect, at a 3-hourly time-scale, even though the katabatic acceleration reaches average values greater than $40 \, \mathrm{m \, s^{-1} \, h^{-1}}$, it cannot be considered as the unique driver of near-surface winds variability. The variability of the near-surface winds in the lowest section of the transect is the result of variability in the intensity of both large-scale and katabatic processes as well as variability in the angle between these two accelerations.

Our momentum budget decomposition study unveils deeper insights into the relationship between the magnitude of different accelerations and their correlation with the total wind speed. It underscores the limitation of assessing the synoptic or katabatic nature of near-surface winds solely by studying the individual magnitudes of accelerations on a 3-hourly time scale. In locations where there is not a single driver of temporal variability, high wind speed events can be synoptic-driven, surface-driven or a combination of both when they act in the same direction.

*Code and data availability.*

All data and scripts developed in this study to compute each momentum budget acceleration for July 2010 are available in https://doi.org/10.5281/zenodo.8315142.

*Author contributions.* C.D., C.Ag. and A.O. designed the study and contributed to the output and observation analyses. C.Ag., C.Am. and C.K. set-up the MAR model for Antarctica with several adaptations. C. Ag. performed model simulations and the MAR PGF diagnostic .



C.D. performed the momentum budget decomposition, assembled observational data, post-processed the data, did the bulk of the analysis and made all the figures. C.D. wrote the first draft, with input from C.Ag. and A.O. All authors contributed to discussions in writing this
paper.

*Competing interests.* The authors declare that they have no conflict of interest.

*Acknowledgements.* This publication was funded by the ANR-JCJC Katabatic project (ANR19-CE01-0020-01) to AJO and NSERC Discovery grant DGERC-2021-00213. The authors appreciate the support of the University of Wisconsin-Madison Automatic Weather Station Program for the data set, data display, and information, NSF grant number 1924730. We acknowledge using data from the CALVA project and
CENECLAM and GLACIOCLIM observatories, http://www-lgge.ujf-grenoble.fr/ christo/calva/. This work is part of the AWACA project that has received funding from the European Research Council (ERC) under the European Union's Horizon 2020 research and innovation programme (Grant agreement No. 951596), and part of the POLARiso project that has received funding from the European Union's Horizon 2020 research and innovation programme under the Marie Skłodowska-Curie grant agreement No 841073. The MAR simulations were performed thanks to granted access to the HPC resources of IDRIS under the allocations 2022-AD010114000 made by GENCI. We acknowledge
the work of Xavier Fettweis (Université de Liège) in developping and mantaining the MAR model.



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
