# Peer review of "Understanding the drivers of near-surface winds in Adélie land, East Antarctica"

_EGUsphere, 2023_

## Referee Comment (RC2)

**Review of "Understanding the drivers of near-surface winds in Adélie Land, East Antarctica", by Davrinche et al. (egusphere-2023-2045)**

**General**

In this clearly-written paper, the authors use output from a run of the MAR model to study the drivers of winds along a well-instrumented transect in East Antarctica. They use a well-established methodology due to van den Broeke and van Lipzig (2003) to decompose the pressure gradient force into large-scale, thermal wind and katabatic components and then examine how variations in each component contribute to the spatial variability of the near-surface wind and its temporal variability from daily to seasonal timescales. Previous studies of winds in this region have focussed on katabatic driving but this study reveals the importance of forcing by large-scale pressure gradients, particularly on shorter timescales. Although the study focuses on a single transect, the results are likely to hold for all of East Antarctica between around 0 and 150°E, where topography and synoptic meteorology are similar to those in the region studied here. The paper is a useful contribution to our understanding of the meteorology and climatology of Antarctica and I recommend that it should be published after minor revision that takes into account the points listed below.

**Specific points**

1. L48: "Interdiurnal" is not very clear. Maybe "...the variability of these winds on daily to monthly time scales"?

2. L65: Replace "…and fast easterlies on the shore" with "…and strong easterlies along the coast".

3. Figure 1: Is the spacing of the dots related to the model grid?

4. L72-73: Do you know why the model correlates better with tower data?

5. L94: "…a horizontal resolution…"

6. L111: Capitalise "Near" at start of second sentence.

7. L129-130: Have you investigated how sensitive your results are to the various parameters used to decompose the potential temperature profile into its "background" and "near-surface" components? Demonstrating that your results are insensitive to the exact choice of, e.g. the height range for calculating $\theta_0$ would add confidence to your findings.

8. Figure 3: Please include the station identifier in the legend for each panel in column (a) to help the reader.

9. L198: "This includes a good representation of the seasonal cycle (Fig. 3b) …". Actually, it looks as if MAR significantly overestimates the annual cycle at the D17 tower.

10. Table 3: You only show magnitudes of the accelerations here. Have you looked at the x- and y-components separately and, in particular, investigated whether (within expected uncertainty) they sum to zero, indicating closure of the momentum budget. Again, this would add confidence to your findings.

11. L271-272: Why have you excluded ADVH from the list of terms studied? From Table 3, it looks as if it is locally and seasonally as important as some other terms.

12. L276: "…spatial standard deviation…". Is this the standard deviation of all transect gridpoints for the mean July profile? Is this a useful metric - I would have thought that the range tells you everything that you want to convey?

13. L277: Maybe "...the product of the surface slope and the potential temperature deficit..."?

14: L284: Maybe "...inland of the coast", rather than "…from the coast" to avoid ambiguity?

15: L288: Figs. 7a and 7b referred to in wrong order.

16: L292-293: Maybe replace "This weaker mean intensity is due to the changing location of synoptic perturbations." with "However, the magnitude of the large-scale acceleration term varies greatly with a changing synoptic situation."

17: L323: "… EITHER the seasonal variability of the total wind speed, OR…"

18: L416: Maybe "might not…" rather than "would not…"?

---

## Author Comment (AC1)

**RESPONSE TO ANONYMOUS REFEREE #2**

REVIEW OF DAVRINCHE ET AL., 2023 – UNDERSTANDING THE DRIVERS OF NEAR-SURFACE WINDS IN ADÉLIE LAND, EAST ANTARCTICA

We thank the reviewer for their valuable and helpful comments on the manuscript. We propose to implement the following changes in a revised version.

Black = reviewer comment / Blue = author's response / *Italic* = revised text.

**1. Specific comments**

L48: "Interdiurnal" is not very clear. Maybe "...the variability of these winds on daily to monthly time scales"?

Yes, we will rephrase in the next revised version: *on sub-daily to monthly timescales*

L65: Replace "...and fast easterlies on the shore" with "...and strong easterlies along the coast".

This will be replaced in the next revision

Figure 1: Is the spacing of the dots related to the model grid?

Yes, it will be stated clearly in the next revised version

L72-73: Do you know why the model correlates better with tower data?

The centre of the grid cell is 14 km from D10 and 13 km from D17. We also know that the model output is closer to D17, because this station is more continental and MAR does not properly represent the oceanic conditions at the coast. Furthermore, we realized that we had made a mistake in the preprocessing of the D10 AWS. It will be corrected in the next revised version (Fig. RR1(c)). For Concordia, the reason is different: the wind sensors on the American tower and the AWS are different. Genthon et al. [2010]. showed that the AWS temperature was biased because the instruments were not ventilated. As a result, people tend to trust the American tower's measurements more, even though it has not been demonstrated that the tower had a better performance for wind's measurements. We acknowledge that we cannot favour one site over another, and will add the AWS data (at least for DC) to Fig. 7c of the manuscript in the next revision.

[Figure]

Figure R1. From top to bottom D17, D47, D85 and Dome C
(a) Comparison of 3-Hourly MAR outputs (black lines) with me-
teorological tower measurements (when available, i.e. at DC and
D17/D10) and AWS (coloured lines). (b) Seasonal cycle computed
for the years available in each AWS, with MAR, AWS and the me-
teorological towers. (c) Scatter plots comparing observations and
model outputs for each station. Black solid lines indicate the y=x
line while the dotted ones are the linear fit associated with each
evaluations. The determination coefficient $R^2$ is indicated next to
each scatter plot.

L94: "...a horizontal resolution..."
This will be corrected in the next revision

L111: Capitalise "Near" at start of second sentence.
This will be replaced in the next revision

L129-130: Have you investigated how sensitive your results are to the various
parameters used to decompose the potential temperature profile into its "back-
ground" and "near-surface" components? Demonstrating that your results are
insensitive to the exact choice of, e.g. the height range for calculating $\theta_0$ would
add confidence to your findings.
Yes. We define the minimum value $H_{min}$ for the interpolation of $\theta_0$ as the height
above which the first vertical derivative of $\theta$ becomes greater than a threshold

(i.e. N*$\gamma_{350-500}$), with N=5. The sensitivity of our interpolation to the value of N is shown in Fig. S1 and S2. Additionally, in the next revision, we will add a sensitivity test to the upper boundary of the height range ($H_{max}$). A comparison with a method based on the second derivative, instead of the first derivative (as suggester by reviewer #1) will also be added to the supplement of the next revised version of the paper.

Figure 3: Please include the station identifier in the legend for each panel in column (a) to help the reader

This will be done in the next revision

L198: "This includes a good representation of the seasonal cycle (Fig. 3b) ...". Actually, it looks as if MAR significantly overestimates the annual cycle at the D17 tower.

Yes, indeed. MAR overestimates the seasonal cycle compared to the D17 tower. The sentence "a good representation of the seasonal cycle" refers more to the spatial differences in the seasonal cycle. The authors wanted to highlight the fact that in both observations and MAR, the seasonal cycle is more pronounced in coastal and lower elevation areas than in the interior. However, it is true that MAR overestimates winter surface winds, leading to an overestimation of the seasonal cycle by almost 60 %. We will state that in the revised version of the manuscript.

You only show magnitudes of the accelerations here. Have you looked at the x- and y- components separately and, in particular, investigated whether (within expected uncertainty) they sum to zero, indicating closure of the momentum budget. Again, this would add confidence to your findings.

The turbulent acceleration term is in fact computed as a residual term separately in the x and y directions. Therefore, the x- and y- components sum to zero, by construction. However, we have confidence in our decomposition, because in both x- and y- direction, Our PGF native output correlates fairly well to our decomposition ($R^2 = 0.8$ in the cross-slope direction and $R^2 = 0.9$ in the downslope direction, see Fig. RR2).

[Figure]

FIGURE R2. Pressure Gradient Force computed using our momentum budget decomposition (MBD) and natively computed by MAR in the (a) Cross-slope direction and (b) downslope direction

[Figure]

FIGURE R3. Map of mean 3-hourly July 2010-2020 horizontal advection (ADVH) and associated wind-vectors

L271-272: Why have you excluded ADVH from the list of terms studied? From Table 3, it looks as if it is locally and seasonally as important as some other terms. ADVH can indeed be important, especially in the slope break. In the next revision, we will add a map of ADVH (Fig. RR3)in Fig.7, plot its annual cycle in Fig. 8 and compute the correlation coefficients of ADVH and WS in Fig.9.

L276: "...spatial standard deviation...". Is this the standard deviation of all transect gridpoints for the mean July profile? Is this a useful metric - I would have thought that the range tells you everything that you want to convey
*The authors thank the reviewer for this comment. It is the standard deviation of all transect grid points for the mean July profile. It doesn't convey any additional information. Thus, in the next revision, this metrics will be removed.*

L277: Maybe "...the product of the surface slope and the potential temperature deficit..."
*This will be corrected in the next revision*

L284: Maybe "...inland of the coast", rather than "...from the coast" to avoid ambiguity
*This will be corrected in the next revision*

L288: Figs. 7a and 7b referred to in wrong order
*This will be corrected in the next revision*

L292-293: Maybe replace "This weaker mean intensity is due to the changing location of synoptic perturbations." with "However, the magnitude of the large-scale acceleration term varies greatly with a changing synoptic situation."
*In the next revision, we will add the suggested sentence and move the original one to the end of the paragraph: 'The magnitude of the large-scale acceleration term varies greatly with a changing synoptic situation. In winter, at D47, for instance, the large-scale acceleration displays a mean value of 5.4 m s$^{-1}$ h$^{-1}$, but a value of the 99$^{th}$ percentile (computed with 3-hourly outputs) of about 12.6 m s$^{-1}$ h$^{-1}$, which is comparable to the mean value of the katabatic acceleration for that period. The weaker mean intensity is due to the changing location of synoptic perturbations.'*

L323: "... EITHER the seasonal variability of the total wind speed, OR...
*This will be corrected in the next revision*

L416: Maybe "might not..." rather than "would not..."
*This will be corrected in the next revision*

**REFERENCES**

Christophe Genthon, Michael S. Town, Delphine Six, Vincent Favier, Stefania Argentini, and Andrea Pellegrini. Meteorological atmospheric boundary layer measurements and ECMWF analyses during summer at Dome C, Antarctica. *Journal of Geophysical Research: Atmospheres*, 115(D5):2009JD012741, March 2010. ISSN 0148-0227. doi: 10.1029/2009JD012741. URL https://agupubs.onlinelibrary.wiley.com/doi/10.1029/2009JD012741.

---

## Author Comment (AC2)

**RESPONSE TO ANONYMOUS REFEREE #1**

REVIEW OF DAVRINCHE ET AL., 2023 – UNDERSTANDING THE DRIVERS OF NEAR-SURFACE WINDS IN ADÉLIE LAND, EAST ANTARCTICA

We thank the reviewer for their valuable and helpful comments on the manuscript. We propose to implement the following changes in a revised version.

Black = reviewer comment / Blue = author's response / *Italic* = revised text.

**1. Major comments**

The method of determining the minimum height above which the potential temperature profile is assumed linear is not explained clearly but seems inaccurate. This is a fundamental aspect of the work presented and will impact all additional results so it is critical that this method is clearly described, accurate and justified. As described starting on line 129 an initial linear vertical potential temperature gradient is estimated between 500 and 350 hPa. I assume that this is done at each 3 h time step anblackd for each model horizontal grid point but this should be stated explicitly in the text.

The interpolation is indeed performed at each 3-hourly time step and for each model horizontal grid point. We will state that explicitly in the text.

The minimum height to be used for the assumed linear potential temperature profile is then based on the height at which the vertical potential temperature gradient exceeds the average gradient between 500 and 350 hPa by a factor of 5. The logic in this seems flawed since what is desired is separating the portion of the profile, near the surface, where the gradient varies with height from further aloft where the gradient is nearly constant with height. Using a constant factor to compare the gradients only determines the height at which the gradient is larger than that in the 500 to 350 hPa layer, which is not the metric that is relevant. In this case what is relevant is assessing how the gradient changes with height - a $2^{nd}$ derivative of potential temperature with height. When this $2^{nd}$ derivative becomes small enough the profile can be assumed to be linear. The authors should consider using this more direct way of assessing the height at which the the potential temperature profile switches from being curved to being linear.

Many thanks for this comment. The method we have used is not straightforward and needs to be better justified. We will develop this point in the revised paper. We considered using a criterium based on the $2^{nd}$ derivative instead of a $1^{st}$ derivative and found that:

- Both of these methods require to chose a threshold and the choice of this threshold is not obvious
- These methods are equivalent and do not introduce significant changes in the final value of $\theta_0$ for all stations (see Fig. RR2)

Mathematically, the $1^{st}$ derivative of a linear curve is a constant (the slope), while its $2^{nd}$ derivative is zero. Thus, in the linear part of the vertical profile of potential temperature, the $1^{st}$ derivative is a constant ($\gamma$) and the $2^{nd}$ derivative is zero. We chose to define the deviation from the linear part as the height ($H_{5*\gamma_{350-500hPa}}$)at which the $1^{st}$ vertical derivative becomes greater than a certain threshold ($Thresh_{350-500hPa}=5*\gamma_{350-500hPa}$). Had we chosen to define the deviation from the linear part as the height ($H_{\frac{\partial^2\theta}{\partial z^2}}$) at which the $2^{nd}$ vertical derivative is no longer equal to zero, we would have had to define a thresshold as well ($Thresh_{\frac{\partial^2\theta}{\partial z^2}}$).

If we use a criterium on the $2^{nd}$ derivative, we have to define a value $Thresh_{\frac{\partial^2\theta}{\partial z^2}}$ for which, as soon as $\frac{\partial^2\theta}{\partial z^2} > Thresh_{\frac{\partial^2\theta}{\partial z^2}}$, we are no longer in the linear part of the potential temperature profile of the atmosphere. The choice of a value $Thresh_{\frac{\partial^2\theta}{\partial z^2}}$ is not obvious.

- The vertical discretization is different close to the ground than higher up in the atmosphere, meaning that there can be some artificial discontinuities in the $2^{nd}$ derivative
- The 2nd derivative in the "linear part" is not exactly zero, because the profile is not perfectly linear. Therefore, one must be carefull to define $Thresh_{\frac{\partial^2\theta}{\partial z^2}}$ big enough, so that it does not result in an artificially high value of $H_{\frac{\partial^2\theta}{\partial z^2}}$.
- $Thresh_{\frac{\partial^2\theta}{\partial z^2}}$ cannot be too big, because otherwise, we might miss the deviation and interpolate too low.
- $Thresh_{\frac{\partial^2\theta}{\partial z^2}}$ must be valid for all 3-hourly time step and grid point

Using a threshold on the $2^{nd}$ derivative requires to find a compromise for the value of $H_{\frac{\partial^2\theta}{\partial z^2}}$ Therefore, a fixed threshold for the $2^{nd}$ derivative did not appear as an easier method than a threshold for the $1^{st}$ derivative.

For the choice of $H_{5*\gamma_{350-500hPa}}$, the initial idea was the following:

Most of the profiles at a 3-hourly time step appeared to be approximately linear in the range of $Z_{350hPa}$ and $Z_{350hPa}$. Therefore, $\gamma_{350-500hPa}$ is a first guess, all over Antarctica of the constant value of $\frac{\partial\theta}{\partial z}$ in the linear part of the vertical profile of $\theta$. It corresponds to the reference value of $\frac{\partial\theta}{\partial z}$ from which $\frac{\partial\theta}{\partial z}$ will deviate under $H_{min}$. To determine $H_{min}$, we need to identify a threshold for the $1^{st}$ derivative.

First option would be to use a fixed threshold (i.e. $Thresh_{\frac{\partial\theta}{\partial z}} = \gamma_{350-500hPa}+\alpha$) with $\alpha$ a constant to be determined. However we realized that for vertical profiles displaying a high $\gamma_{350-500hPa}$, the threshold needed to be higher than for smaller

$\gamma_{350-500hPa}$. Therefore we decided to chose a threshold proportional to $\gamma_{350-500hPa}$ :

$Thresh_{\frac{\partial\theta}{\partial z}} = \gamma_{350-500hPa} + N * \gamma_{350-500hPa}$ with N=4

A sensitivity study of the coefficient N is provided in the supplement.

As a conclusion, both of these methods require to define a threshold, and its definition is not obvious. This will be further discussed and explained in the next version of the manuscript. For the moment, the authors would like to share a first comparison of these methods in Fig. RR1 and Fig. RR2

If the original method will be retained the authors need to better justify this approach by showing a comparison with the more direct method described here. And, the text needs to more explicitly describe the process used to determine this minimum height above which the potential temperature profile is assumed to be linear.

While we decided to stick with the initial method, we understand that there is a concern regarding the robustness of this method. Therefore, we will add in the supplement a comparison of the results of our method with the ones from a method using the $2^{nd}$ derivative. We will also add in the manuscript a more comprehensive description of this approach.

Another concern comes from using 350 hPa as the upper height for the linear approximation of the potential temperature profile. How often is this height above the tropopause. It seems like it would be better to calculate the linear profile over a fixed depth above Hmin - maybe just 100 or 200 hPa - to minimize the possibility of estimating a linear gradient over different layers of the free atmosphere with possibly different air masses and potential temperature gradients.

The authors thank the referee for this comment. We will investigate the sensitivity of our method to the upper height for the linear approximation of the potential temperature. The range of pressure $[350-500hPa]$ was initially chosen because it corresponds to a height at which we were confident that we had approximately a linear response of the potential temperature profile on the Antarctic Plateau (at Concordia station for instance, where the surface pressure is around 650 hPa). However, this might be less accurate in coastal areas. First, we will assess the height of the tropopause and provide a comparison with $H_{max} = Z_{350hPa}$. We will make sure that $H_{max} \leq H_{tropopause}$. However, we want to have $H_{max}$ as high as possible in order to be representative of the whole free troposphere. Furthermore, we want to avoid potential discontinuities related to the vertical discretization that might arise from a low $H_{max}$.

The use of the term thermal wind in your decomposition is confusing. The thermal wind, as defined in atmospheric dynamics text books (e.g. Holton and Hakim) refers to a change in geostrophic wind over some depth of the atmosphere. This is not what this term represents in your decomposition? Parish and Cassano (2003) have the same term in their decomposition and refer to it as the integrated deficit term while Cassano and Parish (2000) referred to this as an adverse pressure gradient force term since it often opposes the downslope flow due to a deepening of the boundary layer with downslope distance and thus a

[Figure]

FIGURE R1. Vertical mean July 2018 profiles of (a, d, g, j) $\theta$, (b, e, h, k) $\frac{\partial \theta}{\partial z}$ and (c, f, i, l) $\frac{\partial^2 \theta}{\partial z^2}$ at D17, D47, 85 and DC (from top to bottom). The blue dotted lines in the middle pannels indicate the minimum height for interpolation of $\theta_0$ computed using the $1^{st}$ order vertical derivative method described in the manuscript. The black dashed lines in the right pannels indicate the minimum height for interpolation of $\theta_0$ computed using a $2^{nd}$ order derivative method for three different values of $Thresh_{\frac{\partial^2 \theta}{\partial z^2}}$

[Figure]

FIGURE R2. $\theta_0$ at surface level computed computed (a) using the method (described in the manuscript) based on the $1^{st}$ order vertical derivative (b) using a method based on the $2^{nd}$ order vertical derivative, with a threshhold $\frac{\partial^2 \theta}{\partial z^2} = 0.0001 K/m^2$ (c) difference between $\theta_0$ computed using method (a) and (b)

larger integrated potential temperature deficit. This term needs to be renamed to more accurately describe what it represents physically.

Here, we define the thermal wind as in Mahrt [1982], van den Broeke and van Lipzig [2003] and. Vihma et al. [2011]. As stated by van den Broeke and van Lipzig [2003]: "It represents the pressure gradient force due to horizontal changes in $\hat{\theta}$". From the geostrophic and hydrostatic equations, the Vallis [2017] textbook defines thermal wind as following:

$\frac{\partial u_{THWD}}{\partial p} = \frac{R}{fP} \nabla_p T$

Which transforms to: $u_{THWD} = \frac{g}{f\theta_0} \nabla_p \int_z^{h_s+h} \theta(z') dz'$

From the textbook indeed, thermal wind represents the pressure gradient force due to horizontal changes in the vertically integrated potential temperature and not the vertically integrated potential temperature deficit. It is true that it might be confusing for the reader to call it "Thermal wind". We propose to rename it $THWD_{TD}$ where TD stands for "temperature deficit". We will mention in the manuscript the different names found in the literature for this term and state more clearly that it does not correspond to a proper thermal wind.

Figure 4: There are obvious discontinuities in the pressure gradient force components (e.g. KAT and THWD between D17 and D85) seen in this figure. The source of these clearly non-physical results need to be discussed. Do these artifacts reflect a shortcoming in the decomposition that makes the results less trustworthy?

While it is true that there exist sharp gradients between D17 and D85 in the katabatic term, the authors would like to underline that these sharp gradients do not originate from the decomposition itself, but rather from the multiplication by the sinus of the slope. This is why we have plotted the vertical profile of the

potential temperature deficit $\Delta$ next to the vertical profile of the katabatic accel-eration. We do not have any sharp discontinuity in the profile of $\Delta$. Regarding the thermal wind, we have plotted $\hat{\theta}$ next to the THWD acceleration. There is no discontinuity in the vertical profile of $\hat{\theta}$, but there is a strong minimum at the foot of the slope (next to D17), where turbulence and advection create a strong mixing of the boundary layer which reduces $\hat{\theta}$. Therefore, this zone corresponds to the sharpest horizontal gradient of $\hat{\theta}$.

Figure 7: I found that showing the direction of the momentum budget terms as the equivalent geostrophic wind to be confusing. It would be clearer to simply show vectors in the direction of each momentum budget term scaled by their magnitude. In this way it will be clear in which direction each force is acting rather than the reader needing to rotate the vectors mentally by 90 deg. If the authors wish to keep the vectors scaled relative to a geostrophic wind speed the magnitude of each term can simply be divided by the Coriolis parameter, which will retain the same magnitude as currently shown in Figure 7 but without the direction being rotated 90 deg from the true direction each force is acting.

In this figure, we wanted to show the direction of the resulting wind speed associated with each accelerations in the quasi geostrophic hypothesis. We have demonstrated that this hypothesis is valid in Section 4.1. With this hypothesis, the resulting wind speed associated for instance with the katabatic acceleration is rotated by 90 and the total wind-speed is the sum of the rotated wind com-ponents. We thought that the readers would rather like to see the wind-direction than the direction of the accelerations themselves. For example, they would ex-pect the large-scale winds on the ocean to be westerlies. We also wanted to underpin the cross-slope direction of the katabatic winds, that are sometimes wrongly assumed to blow downslope. We would like to keep on showing the wind vectors, but we will make sure to add a better description for more clarity.

**2. Minor comments**

Line 14: remove latitudes after sub polar - it is redundant and not needed
This will be corrected in the next revision

Figure 1 caption: Last sentence of caption describing color of dots does not match what is shown in the figure.
This will be corrected in the next revision

Line 69: model should be model's
This will be corrected in the next revision

Line 73: What is meant by "the data are slightly better correlated to our model"? This sounds like you are selecting observational data that matches the model which is not appropriate - you cannot preferentially choose observations that match your model and ignore and de-emphasize those that don't.

[Figure]

FIGURE R3. From top to bottom D17, D47, D85 and Dome C
(a) Comparison of 3-Hourly MAR outputs (black lines) with me-
teorological tower measurements (when available, i.e. at DC and
D17/D10) and AWS (coloured lines). (b) Seasonal cycle computed
for the years available in each AWS (see Table **??**), with MAR,
AWS and the meteorological towers. (c) Scatter plots comparing
observations and model outputs for each station. Black solid lines
indicate the y=x line while the dotted ones are the linear fit asso-
ciated with each evaluations. The determination coefficient $R^2$ is
indicated next to each scatter plot.

This is a poor choice of word. MAR's grid cell includes both D10 and D17. The
centre of the grid cell is 14 km from D10 and 13 km from D17. The model output
is closer to D17, because this station is more continental. We know that MAR
does not properly represent the oceanic conditions at the coast. Furthermore, we
realized that we had made a mistake in the preprocessing of the D10 AWS. It will
be corrected in the next revised version (Fig. RR3(c)). For Concordia, the reason
is different: the wind sensors on the American tower and the AWS are different.
Genthon et al. [2010]. showed that the AWS temperature was biased because the
instruments were not ventilated. As a result, people tend to trust the American
tower's measurements more, even though it has not been demonstrated that the
tower had a better performance for wind's measurements. We acknowledge that
we cannot favour one site over another, and will add the AWS data (at least for
DC) to Fig. 7c of the manuscript in the next revision.

Table 1: List lon, lat and elevation for DC-tower. I assume that this is the

same as for the DC-AWS but this should be confirmed by listing these values in the table.
Yes, this will be added in the next revision

Line 86: What is meant by model bases? Please clarify.
The authors are referring to the equations of the atmospheric model, the lateral boundary conditions, the upper and lower boundary conditions and the main parametrizations

Lines 93-94: As written it seems like the 30 snow/ice layers are each 20 m thick. I think what you mean is that the total depth of snow/ice is 20 m and there are 30 layers distributed over this depth. Please rephrase.
Yes, this will be corrected in the next revision

Table 2: It would be more informative if the table listed the start and end distance from the coast for each section and gave the range of terrain slope in addition to the average slope.
Yes, this will be done in the next revision

Line 111: winds variability should be wind variability and near-surface should be capitalized since it is at the start of new sentence.
Yes, this will be corrected in the next revision

Line 132: larger that should larger than
Yes, this will be corrected in the next revision

Line 192: Delete boundary between surface and layer. I think you are referring simply to the surface layer here.
Yes, this will be corrected in the next revision

Table 3: It would be helpful if the relative magnitude of each term was given. For example, the terms could be normalized relative to the LSC term or total PGF to indicate how much larger or smaller each term is relative to the LSC or overall PGF forcing. This could be given as a percentage in parenthesis after the seasonal value is listed. The total PGF should also be listed in this table.
We will add the percentage of the PGF for each accelerations

Figure 7: It would be helpful to add a panel showing the total pressure gradient force, which can then be compared to the other terms in the momentum equation.
We will replace pannel 7.d by the total PGF

Figure 7: Similar to the comment regarding Table 3, showing figures of the ratio of KAT, THWD and TURB to LSC would be very helpful, especially if a color bar with different colors above (forcing greater than LSC) and below (forcing less than LSC) was used. This would clearly show where each forcing term exceeds

the LSC forcing.
We will add this figure in the supplement

Line 330: Replace outputs with output
Yes, this will be corrected in the next revision

**References**

L. Mahrt. Momentum balance of gravity flows. 1982.

M. R. van den Broeke and N. P. M. van Lipzig. Factors Controlling the Near-Surface Wind Field in Antarctica*. *Monthly Weather Review*, 131(4):733–743, April 2003. ISSN 0027-0644, 1520-0493. doi: 10.1175/1520-0493(2003) 131⟨0733:FCTNSW⟩2.0.CO;2. URL http://journals.ametsoc.org/doi/10.1175/1520-0493(2003)131<0733:FCTNSW>2.0.CO;2.

Timo Vihma, Eveliina Tuovinen, and Hannu Savijärvi. Interaction of katabatic winds and near-surface temperatures in the Antarctic: KATABATIC WINDS IN THE ANTARCTIC. *Journal of Geophysical Research: Atmospheres*, 116 (D21), November 2011. ISSN 01480227. doi: 10.1029/2010JD014917. URL http://doi.wiley.com/10.1029/2010JD014917.

Geoffrey K. Vallis. *Atmospheric and Oceanic Fluid Dynamics: Fundamentals and Large-Scale Circulation*. Cambridge University Press, 2 edition, June 2017. ISBN 978-1-107-06550-5 978-1-107-58841-7. doi: 10.1017/9781107588417. URL https://www.cambridge.org/core/product/identifier/9781107588417/type/book.

Christophe Genthon, Michael S. Town, Delphine Six, Vincent Favier, Stefania Argentini, and Andrea Pellegrini. Meteorological atmospheric boundary layer measurements and ECMWF analyses during summer at Dome C, Antarctica. *Journal of Geophysical Research: Atmospheres*, 115(D5):2009JD012741, March 2010. ISSN 0148-0227. doi: 10.1029/2009JD012741. URL https://agupubs.onlinelibrary.wiley.com/doi/10.1029/2009JD012741.

---

## Author Response (AR1)

**RESPONSE TO REVIEWERS**

REVIEW OF DAVRINCHE ET AL., 2023 – UNDERSTANDING THE DRIVERS OF NEAR-SURFACE WINDS IN ADÉLIE LAND, EAST ANTARCTICA

**We thank the reviewers for their time and their valuable and helpful comments on the manuscript. We have implemented the following changes in a revised version.**

Black = reviewer comment / Blue = author's comment / *Italic* = revised text.
* * *
**1. RESPONSE TO REVIEWER 1**

The method of determining the minimum height above which the potential temperature profile is assumed linear is not explained clearly but seems inaccurate. This is a fundamental aspect of the work presented and will impact all additional results so it is critical that this method is clearly described, accurate and justified. As described starting on line 129 an initial linear vertical potential temperature gradient is estimated between 500 and 350 hPa. I assume that this is done at each 3 h time step anblackd for each model horizontal grid point but this should be stated explicitly in the text.

The interpolation is indeed performed at each 3-hourly time step and for each model grid cell. We have added the following sentence line 125: *These definitions are based on the hypothesis that we can define for each grid-cell and each time-step a minimum height $H_{min}$ above which the vertical profile of $\theta$ is quasi-linear, and the free atmosphere is not influenced by surface processes.*

The vertical potential temperature gradient is then calculated and compared to the gradient between 500 and 350 hPa. I assume that this new gradient is calculated at each model vertical level moving up from the surface but this is not stated explicitly and needs to be.

Yes, the vertical potential temperature gradient is calculated at each level, moving up from the surface. We have added the following sentence line 135: *we look for the minimum height $H_{min}$ under which the vertical derivative of potential temperature computed at each level deviates from $\gamma_{350-500}$.*

The minimum height to be used for the assumed linear potential temperature profile is then based on the height at which the vertical potential temperature gradient exceeds the average gradient between 500 and 350 hPa by a factor of 5.

The logic in this seems flawed since what is desired is separating the portion of
the profile, near the surface, where the gradient varies with height from further
aloft where the gradient is nearly constant with height. Using a constant factor to
compare the gradients only determines the height at which the gradient is larger
than that in the 500 to 350 hPa layer, which is not the metric that is relevant.

In this case what is relevant is assessing how the gradient changes with height
- a $2^{nd}$ derivative of potential temperature with height. When this $2^{nd}$ deriva-
tive becomes small enough the profile can be assumed to be linear. The authors
should consider using this more direct way of assessing the height at which the
the potential temperature profile switches from being curved to being linear.

Many thanks for this comment. The method we have used is not straightfor-
ward and needed to be better justified. First, we considered using a criterion
based on the $2^{nd}$ derivative instead of a $1^{st}$ one and found that:

- Both of these methods require to choose a threshold and the choice of this
  threshold is not obvious
- These methods are equivalent and do not introduce significant changes in
  the final value of $\theta_0$ for all stations (see Fig. S4)

In the linear part of the vertical profile of potential temperature, the $1^{st}$ vertical
derivative is a constant $(\gamma)$ and the $2^{nd}$ derivative is zero. We chose to define the
deviation from the linear part as the height at which the $1^{st}$ vertical derivative
deviates from the constant value in the linear part $(\approx \gamma_{350-500hPa})$ by more than
a certain threshold $(|\frac{\partial\theta}{\partial z} - \gamma_{350-500hPa}| > Thresh_{\frac{\partial\theta}{\partial z}})$.

Had we chosen to define the deviation from the linear part as the height $(H_{\frac{\partial^2\theta}{\partial z^2}})$
at which the $2^{nd}$ vertical derivative is no longer equal to zero, we would have had
to define a threshold as well $(Thresh_{\frac{\partial^2\theta}{\partial z^2}})$.

In this case, if $|\frac{\partial^2\theta}{\partial z^2}| > Thresh_{\frac{\partial^2\theta}{\partial z^2}}$, we are no longer in the linear part of the
potential temperature profile of the atmosphere. The choice of a value $Thresh_{\frac{\partial^2\theta}{\partial z^2}}$
is not obvious because:

- The vertical discretization is different close to the ground than higher up in
  the atmosphere, meaning that there can be some artificial discontinuities
  in the $2^{nd}$ derivative
- The $2^{nd}$ derivative in the "linear part" is not exactly zero, because the
  profile is not perfectly linear. Therefore, one must be careful to define
  $Thresh_{\frac{\partial^2\theta}{\partial z^2}}$ large enough, so that it does not result in an artificially high
  value of $H_{\frac{\partial^2\theta}{\partial z^2}}$.
- $Thresh_{\frac{\partial^2\theta}{\partial z^2}}$ cannot be too large, because otherwise, we might miss the
  deviation and interpolate too low.
- $Thresh_{\frac{\partial^2\theta}{\partial z^2}}$ must be valid for all 3-hourly time step and grid point

Using a threshold on the $2^{nd}$ derivative requires to find a compromise for the value of $H_{\frac{\partial^2 \theta}{\partial z^2}}$ Therefore, a fixed threshold for the $2^{nd}$ derivative did not appear as an easier method than a threshold for the $1^{st}$ derivative.

For the choice of $Thresh_{\frac{\partial \theta}{\partial z}} = 4 * \gamma_{350-500hPa}$, the initial idea was the following:
Most of the profiles at a 3-hourly time step appeared to be approximately linear in the range of $Z_{350hPa}$ to $Z_{500hPa}$. Therefore, $\gamma_{350-500hPa}$ is a first guess, all over Antarctica, of the constant value of $\frac{\partial \theta}{\partial z}$ in the linear part of the vertical profile of $\theta$. It corresponds to the reference value of $\frac{\partial \theta}{\partial z}$ from which $\frac{\partial \theta}{\partial z}$ will deviate under $H_{min}$. To determine $H_{min}$, we need to identify a threshold for the $1^{st}$ derivative.
We have added in the manuscript line 138 that: *A first option would be to determine a constant threshold in time and space. However we realised that for vertical profiles with a high $\gamma_{350-500}$, the threshold needed to be higher than for smaller $\gamma_{350-500}$. Therefore we decided to choose a threshold proportional to $\gamma_{350-500}$.*
$Thresh_{\frac{\partial \theta}{\partial z}} = N \cdot \gamma_{350-500hPa}$ with N=4
A sensitivity study of the coefficient N is provided in the supplement.
As a conclusion, both of these methods require to define a threshold, and this definition is not obvious.

If the original method will be retained the authors need to better justify this approach by showing a comparison with the more direct method described here. And, the text needs to more explicitly describe the process used to determine this minimum height above which the potential temperature profile is assumed to be linear.
While we decided to stick with the initial method, we understand that there is a concern regarding the robustness of this method. Therefore, we have added a comparison of the results of our method with the ones from a method using the $2^{nd}$ derivative (Fig. S3 and Fig. S4 in the supplement, also shown hereafter). We have rewritten section 2.2.1 to describe better our method.
Another concern comes from using 350 hPa as the upper height for the linear approximation of the potential temperature profile. How often is this height above the tropopause. It seems like it would be better to calculate the linear profile over a fixed depth above Hmin - maybe just 100 or 200 hPa - to minimize the possibility of estimating a linear gradient over different layers of the free atmosphere with possibly different air masses and potential temperature gradients.
The authors thank the referee for this comment. We have stated in the manuscript that : *We are confident that pressure levels between 500 hPa and 350 hPa fall within the free troposphere in Antarctica, as the tropopause is typically between 150 hPa and 320 hPa in this region [Hoffmann and Spang, 2022]. Therefore, the slope of the linear interpolation of $\theta$ between 500 hPa and 350 hPa ($\gamma_{350-500hPa}$) gives a good first estimate of $\gamma_0$. We want to have $H_{max}$ as high as possible in order to be representative of the whole free troposphere. Furthermore, we want to avoid potential horizontal discontinuities. As MAR vertical levels are discretized using sigma coordinates, up in the troposphere, vertical levels are more spaced and we must be extremely careful not to introduce jumps between*

[Figure]

FIGURE S3. Vertical mean July 2018 profiles of (a, d, g, j) $\theta$, (b, e, h, k) $\frac{\partial \theta}{\partial z}$ and (c, f, i, l) $\frac{\partial^2 \theta}{\partial z^2}$ at D17, D47, 85 and DC (from top to bottom). The blue dotted lines in the middle pannels indicate the minimum height for interpolation of $\theta_0$ computed using the $1^{st}$ order vertical derivative method described in the manuscript. The black dashed lines in the right pannels indicate the minimum height for interpolation of $\theta_0$ computed using a $2^{nd}$ order derivative method for three different values of $Thresh_{\frac{\partial^2 \theta}{\partial z^2}}$

[Figure]

FIGURE S4. $\theta_0$ at surface level computed computed (a) using the method (described in the manuscript) based on the $1^{st}$ order vertical derivative (b) using a method based on the $2^{nd}$ order vertical derivative, with a threshold $\frac{\partial^2 \theta}{\partial z^2} = 0.0001 K/m^2$ (c) difference between $\theta_0$ computed using method (a) and (b)

two neighboring grid points. By selecting $H_{max}$ as the closest vertical level to $H_{min} + 200hPa$, two neighboring grid points might have significantly different values of $H_{max}$. Therefore, we decided to stick with the original method and to use $H_{max} = 500hPa$.

The use of the term thermal wind in your decomposition is confusing. The thermal wind, as defined in atmospheric dynamics text books (e.g. Holton and Hakim) refers to a change in geostrophic wind over some depth of the atmosphere. This is not what this term represents in your decomposition? Parish and Cassano (2003) have the same term in their decomposition and refer to it as the integrated deficit term while Cassano and Parish (2000) referred to this as an adverse pressure gradient force term since it often opposes the downslope flow due to a deepening of the boundary layer with downslope distance and thus a larger integrated potential temperature deficit. This term needs to be renamed to more accurately describe what it represents physically.

Thank you for this comment. As stated by van den Broeke and van Lipzig [2003]: "It (thermal wind) represents the pressure gradient force due to horizontal changes in $\hat{\theta}$". From the geostrophic and hydrostatic equations, the Vallis [2017] textbook defines thermal wind as following:

$\frac{\partial u_{THWD}}{\partial p} = \frac{R}{fP}\nabla_p T$

Which transforms to: $u_{THWD} = \frac{g}{f\theta_0}\nabla_p \int_z^{h_s+h} \theta(z')dz'$

From the textbook indeed, thermal wind represents the pressure gradient force due to horizontal changes in the vertically integrated potential temperature and not the vertically integrated potential temperature deficit. It is true that it might be confusing for the reader to call it "Thermal wind". Therefore, we have renamed it $THWD_{TD}$ where TD stands for "temperature deficit". We have also written the following paragraph in the revised manuscript : *The thermal wind acceleration (**THWD**$_{TD}$) is a function of the horizontal gradients of $\hat{\theta}$, the vertically*

*integrated potential temperature deficit between the ground and $z_{max}$ (Equation (4) and Fig. 2). Note that the classic definition of thermal wind does not include a vertically integrated gradient of potential temperature deficit but of potential temperature. Here, we use the definition of van den Broeke and van Lipzig [2003] while Parish and Cassano [2003] named this term "integrated deficit".*

Figure 4: There are obvious discontinuities in the pressure gradient force components (e.g. KAT and THWD between D17 and D85) seen in this figure. The source of these clearly non-physical results need to be discussed. Do these artifacts reflect a shortcoming in the decomposition that makes the results less trustworthy?

While it is true that there exist sharp gradients between D17 and D85 in the katabatic term, the authors would like to underline that these sharp gradients do not originate from the decomposition itself, but rather from the multiplication by the sinus of the slope. This is why we have plotted the vertical profile of the potential temperature deficit $\Delta$ next to the vertical profile of the katabatic acceleration. We do not have any sharp discontinuity in the profile of $\Delta$. Regarding the thermal wind, we have plotted $\hat{\theta}$ next to the THWD acceleration. There is no discontinuity in the vertical profile of $\hat{\theta}$, but there is a strong minimum at the foot of the slope (next to D17), where turbulence and advection create a strong mixing of the boundary layer which reduces $\hat{\theta}$. Therefore, this zone corresponds to the sharpest horizontal gradient of $\hat{\theta}$.

Figure 7: I found that showing the direction of the momentum budget terms as the equivalent geostrophic wind to be confusing. It would be clearer to simply show vectors in the direction of each momentum budget term scaled by their magnitude. In this way it will be clear in which direction each force is acting rather than the reader needing to rotate the vectors mentally by 90 deg. If the authors wish to keep the vectors scaled relative to a geostrophic wind speed the magnitude of each term can simply be divided by the Coriolis parameter, which will retain the same magnitude as currently shown in Figure 7 but without the direction being rotated 90 deg from the true direction each force is acting.

In this figure, we wanted to show the direction of the resulting wind speed associated with each accelerations in the quasi geostrophic hypothesis. We have demonstrated that this hypothesis is valid in Section 4.1. With this hypothesis, the resulting wind speed associated for instance with the katabatic acceleration is rotated by 90 ° and the total wind-speed is the sum of the rotated wind components. We thought that the readers would rather like to see the wind-direction than the direction of the accelerations themselves. For example, they would expect the large-scale winds on the ocean to be westerlies. We also wanted to underpin the cross-slope direction of the katabatic winds, that are sometimes wrongly assumed to blow downslope. Therefore, we have decided to keep on showing the wind vectors. Nevertheless, we have added a few sentences to explain that. For example line 302: *Here, we want to emphasise that the katabatic acceleration*

*points in the slope direction. Consequently, in the quasi-geostrophic stationary conditions detailed in Section 4.1, it increases the wind speed in the cross-slope direction, along the elevation contours (Fig. 7b). Therefore, wind vectors associated with the katabatic acceleration are always directed in the cross-slope direction.*

Line 14: remove latitudes after sub polar - it is redundant and not needed
This has been removed.

Figure 1 caption: Last sentence of caption describing color of dots does not match what is shown in the figure.
This has been corrected.

Line 69: model should be model's
This has been corrected

Line 73: What is meant by "the data are slightly better correlated to our model"? This sounds like you are selecting observational data that matches the model which is not appropriate - you cannot preferentially choose observations that match your model and ignore and de-emphasize those that don't.
This was a poor choice of word. We have added the following paragraph line 208-214: *At the coast, the D10 AWS (≈ 3 km from the coast) and D17 weather profiling tower (≈ 10 km from the coast) are contained within the same MAR grid cell, whose centre is equidistant from both stations. MAR correlates slightly better with the observations from D17 (R = 0.61 ) than from D10 (R = 0.53), and both stations are well correlated (R=0.87). This may be due to the fact the model grid cell is more representative of continental than oceanic conditions. The two wind sensors of the American tower and the AWS at Dome C are also located within the same MAR grid cell. Although it has been demonstrated that the AWS temperature was biased because the instruments were not ventilated [Genthon et al., 2010], there has been no assessment of the comparative performance of the wind measurements.* We acknowledge that we cannot favour one site over another, and we have added the AWS data to Fig. 7c in the revised version. Furthermore, we realized that we had made a mistake in the preprocessing of the D10 AWS. It has been corrected in the revised version (Fig. Fig. 3(c), reproduced hereafter)

Table 1: List lon, lat and elevation for DC-tower. I assume that this is the same as for the DC-AWS but this should be confirmed by listing these values in the table.
Yes, this has been added in the revision version.

Line 86: What is meant by model bases? Please clarify.
The authors are referring to the equations of the atmospheric model, the lateral boundary conditions, the upper and lower boundary conditions and the main parametrizations. This list has been added line 87.

Lines 93-94: As written it seems like the 30 snow/ice layers are each 20 m thick. I think what you mean is that the total depth of snow/ice is 20 m and there are

[Figure]

FIGURE 3. From top to bottom D17, D47, D85 and Dome C (a) Comparison of 3-Hourly MAR outputs (black lines) with meteorological tower measurements (when available, i.e. at DC and D17/D10) and AWS (coloured lines). (b) Seasonal cycle computed for the years available in each AWS, with MAR, AWS and the meteorological towers. (c) Scatter plots comparing observations and model outputs for each station. Black solid lines indicate the y=x line while the dotted ones are the linear fit associated with each evaluations. The determination coefficient $R^2$ is indicated next to each scatter plot.

30 layers distributed over this depth. Please rephrase.
Yes, this has been rephrased line 95.

Table 2: It would be more informative if the table listed the start and end distance from the coast for each section and gave the range of terrain slope in addition to the average slope.
Yes, the start and end distance from the coast for each section and the range of terrain slope have been added in Table 2.

Line 111: winds variability should be wind variability and near-surface should be capitalized since it is at the start of new sentence.
Yes, this has been corrected.

Line 132: larger that should larger than
Yes, this has been corrected.

Line 192: Delete boundary between surface and layer. I think you are referring simply to the surface layer here.
Yes, this has been corrected.

Table 3: It would be helpful if the relative magnitude of each term was given. For example, the terms could be normalized relative to the LSC term or total PGF to indicate how much larger or smaller each term is relative to the LSC or overall PGF forcing. This could be given as a percentage in parenthesis after the seasonal value is listed. The total PGF should also be listed in this table.
We have decided not to normalize by the large-scale acceleration. However, we have added black asterisks to highlight the accelerations displaying the highest values for each stations.

Figure 7: It would be helpful to add a panel showing the total pressure gradient force, which can then be compared to the other terms in the momentum equation.
We have added a map of the total PGF.

Figure 7: Similar to the comment regarding Table 3, showing figures of the ratio of KAT, THWD and TURB to LSC would be very helpful, especially if a color bar with different colors above (forcing greater than LSC) and below (forcing less than LSC) was used. This would clearly show where each forcing term exceeds the LSC forcing.
As we show on the same map the mean vectors of each corresponding wind speed, normalizing by the LSC acceleration or the PGF would make it harder for the reader to understand the link between the acceleration and resulting wind speed. Therefore, we thought that it would be clearer to show the absolute value of each accelerations.

Line 330: Replace outputs with output
Yes, this has been corrected.

**2. Response to reviewer 2**

L48: "Interdiurnal" is not very clear. Maybe "...the variability of these winds on daily to monthly time scales"?
Yes, it has been rephrased in the revised version: *on sub-daily to monthly time-scales*

L65: Replace "...and fast easterlies on the shore" with "...and strong easterlies along the coast".
This has been replaced in the revised version.

Figure 1: Is the spacing of the dots related to the model grid?
Yes, it has been stated clearly in the legend of Fig. 1b in the revised version:
*Elevation profile along the transect extracted on the 35-km MAR grid*

L72-73: Do you know why the model correlates better with tower data?
We have added a complimentary information line 208-214: *At the coast, the D10
AWS ($\approx$ 3 km from the coast) and D17 weather profiling tower ($\approx$ 10 km from
the coast) are contained within the same MAR grid cell, whose centre is equidis-
tant from both stations. MAR correlates slightly better with the observations from
D17 (R = 0.61) than from D10 (R = 0.53), and both stations are well correlated
(R=0.87). This may be due to the fact the model grid cell is more representative
of continental than oceanic conditions. The two wind sensors of the American
tower and the AWS at Dome C are also located within the same MAR grid cell.
Although it has been demonstrated that the AWS temperature was biased because
the instruments were not ventilated [Genthon et al., 2010], there has been no
assessment of the comparative performance of the wind measurements.*
    Furthermore, we realized that we had made a mistake in the preprocessing
of the D10 AWS. It has been corrected in the revised version (Fig. 3(c)). We
acknowledge that we cannot favour one site over another, and we have added the
AWS data to Fig. 7c of the manuscript in the next revision.

L94: "...a horizontal resolution..."
This has been corrected in the revised version.

L111: Capitalise "Near" at start of second sentence.
This has been corrected in the revised version.

L129-130: Have you investigated how sensitive your results are to the various parameters used to decompose the potential temperature profile into its "background" and "near-surface" components? Demonstrating that your results are insensitive to the exact choice of, e.g. the height range for calculating $\theta_0$ would add confidence to your findings.
Yes. We define the minimum value $H_{min}$ for the interpolation of $\theta_0$ as the height above which the first vertical derivative of $\theta$ deviates from the constant value ($\gamma_{350-500}$) by more than a threshold (i.e. N·$\gamma_{350-500}$, with N=4). The sensitivity of our interpolation to the value of N is shown in Fig. S1 and S2. Additionally, in the revised version, we have added a comparison with a method based on the second derivative (Fig. S3 and Fig. S4), instead of the first derivative (as suggester by reviewer #1).

Figure 3: Please include the station identifier in the legend for each panel in column (a) to help the reader
This has been done in the revised version.

L198: "This includes a good representation of the seasonal cycle (Fig. 3b) ...".
Actually, it looks as if MAR significantly overestimates the annual cycle at the D17 tower.
Yes, indeed. MAR overestimates the seasonal cycle compared to the D17 tower. The sentence "a good representation of the seasonal cycle" refers more to the spatial differences in the seasonal cycle. The authors wanted to highlight the fact that in both observations and MAR, the seasonal cycle is more pronounced in coastal and lower elevation areas than in the interior. However, it is true that MAR overestimates winter surface winds, leading to an overestimation of the seasonal cycle by almost 60 %. We have stated that in the revised version: *However, across all the other stations, the model tends to overestimate the mean wind speed with a bias ranging from 0.6 m s$^{-1}$ for D85 to 2.0 m s$^{-1}$ at D17. The largest biases are found during winter time at D17 and DC, with an overestimation of the seasonal cycle in MAR, compared to AWS measurements of about 60 % in D17 and 90 % in DC. The strongest correlations are found at sites with higher mean wind speeds such as D47 ($R^2 = 0.7$) and D17 ($R^2 = 0.61$)*
You only show magnitudes of the accelerations here. Have you looked at the x- and y- components separately and, in particular, investigated whether (within expected uncertainty) they sum to zero, indicating closure of the momentum budget. Again, this would add confidence to your findings.
The turbulent acceleration term is in fact computed as a residual term separately in the x and y directions. Therefore, the x- and y- components sum to zero, by

[Figure]

FIGURE R3. Pressure Gradient Force computed using our momentum budget decomposition (MBD) and natively computed by MAR in the (a) Cross-slope direction and (b) downslope direction

construction. However, we have confidence in our decomposition, because in both x- and y- direction, our native PGF output correlates fairly well to our decomposition ($R^2 = 0.8$ in the cross-slope direction and $R^2 = 0.9$ in the downslope direction, see Fig. RR3).

L271-272: Why have you excluded ADVH from the list of terms studied? From Table 3, it looks as if it is locally and seasonally as important as some other terms.
ADVH can indeed be important, especially in the slope break. In the revised paper, we have added a map of ADVH (Fig. 7d), plotted its annual cycle in Fig. 8 and computed the correlation coefficients of ADVH and WS in Fig. 9 and Fig. 10. All these figures are shown hereafter.

We have also added a sentence about the importance of ADVH at specific location (line 187): *The effect of advection cannot fully explain strong correlations between katabatic acceleration and total wind speed. Areas of strong correlation correspond either to locations of strong negative and positive advection contribution ((I), (II), (III)) or weak contribution ((IV), (V), (VI), (VII)).*

[Figure]

FIGURE 7. (Upper- and middle panels) Mean July 2010-2020 norm
of accelerations at surface level (∼7 m a.g.l.) computed with 3-
hourly MAR outputs:(a) large-scale, (b) katabatic, (c) thermal
wind, (d) horizontal advection, (e) turbulence and (f) Pressure
Gradient Force. Superimposed are the equivalent wind vectors.
(Lower panels) Mean July 2010-2020 values of (g) the background
temperature $\theta_0$, (h) the potential temperature deficit $\Delta_\theta$ and (i)
the vertically integrated potential temperature deficit $\hat{\theta}$ at surface
level (∼7 m a.g.l.) computed with 3-hourly MAR outputs.

L276: "...spatial standard deviation...". Is this the standard deviation of all
transect gridpoints for the mean July profile? Is this a useful metric - I would
have thought that the range tells you everything that you want to convey
The authors thank the reviewer for this comment. It is the standard deviation of
all transect grid points for the mean July profile. It doesn't convey any additional
information. Thus, in the revised version, this metric has been removed.

[Figure]

FIGURE 8. Seasonal cycle of 3-hourly winds averaged over 10 years for (a) total wind speed, (b) wind speed equivalent to large-scale acceleration, (c) wind speed equivalent to thermal wind, (d) wind speed equivalent to advection, (e) wind speed equivalent to horizontal katabatic and (f) wind speed equivalent to turbulent accelerations. Note that the y-axis is different between the top panel ($|WS|$, $|V_{LSC}|$, $|V_{THWD_{TD}}|$) and the bottom panel ($|V_{KAT}|$, $|V_{TURB}|$).

[Figure]

FIGURE 9. Correlation coefficient (R) between the 3-hourly total wind speed and the the different accelerations in July 2010-2020.

[Figure]

FIGURE 10. (a) Average July 2010-2020 correlation coefficient of
3-hourly katabatic acceleration and wind speed (b) Average July
2010-2020 correlation coefficient of 3-hourly large-scale acceleration
and wind speed (c) directional constancy of 3-hourly large-scale
wind speed. (d, e, f): Mean of 3-hourly July 2010-2020 scalar
product normalised by the norm of wind speed of (d) 3-hourly
katabatic wind speed and total wind speed, (e) 3-hourly large-scale
and total wind speed, (f) 3-hourly thermal-wind and total wind
speed, (g) 3-hourly advection and total wind speed. For the 7
panels, the dotted black line corresponds to the line for which the
correlation coefficient of katabatic acceleration and total wind speed
reaches 0.5. Seven zones of higher correlations are indicated: (I),
(II), (III), (IV), (V), (VI) and (VII))

L277: Maybe "...the product of the surface slope and the potential tempera-
ture deficit..."
This has been corrected in the revised version.

L284: Maybe "...inland of the coast", rather than "...from the coast" to avoid
ambiguity
This has been corrected in the revised version.

L288: Figs. 7a and 7b referred to in wrong order
This has been corrected in the revised version.

L292-293: Maybe replace "This weaker mean intensity is due to the changing
location of synoptic perturbations." with "However, the magnitude of the large-
scale acceleration term varies greatly with a changing synoptic situation."
In the revised version, we have added the suggested sentence and moved the orig-
inal one to the end of the paragraph (line 315): *The magnitude of the large-scale*

*acceleration term varies greatly with a changing synoptic situation. In winter, at D47, for instance, the large-scale acceleration displays a mean value of 5.4 $m\,s^{-1}\,h^{-1}$, but a value of the $99^{th}$ percentile (computed with 3-hourly outputs) of about 12.6 $m\,s^{-1}h^{-1}$, which is comparable to the mean value of the katabatic acceleration for that period. The weaker mean intensity is due to the changing location of synoptic perturbations.*

L323: "... EITHER the seasonal variability of the total wind speed, OR...
This has been corrected in the revised version.

L416: Maybe "might not..." rather than "would not..."
This has been corrected in the revised version.

**References**

Lars Hoffmann and Reinhold Spang. An assessment of tropopause characteristics of the ERA5 and ERA-Interim meteorological reanalyses. *Atmospheric Chemistry and Physics*, 22(6):4019–4046, March 2022. ISSN 1680-7324. doi: 10.5194/acp-22-4019-2022. URL https://acp.copernicus.org/articles/22/4019/2022/.

M. R. van den Broeke and N. P. M. van Lipzig. Factors Controlling the Near-Surface Wind Field in Antarctica*. *Monthly Weather Review*, 131(4):733–743, April 2003. ISSN 0027-0644, 1520-0493. doi: 10.1175/1520-0493(2003)131⟨0733:FCTNSW⟩2.0.CO;2. URL http://journals.ametsoc.org/doi/10.1175/1520-0493(2003)131<0733:FCTNSW>2.0.CO;2.

Geoffrey K Vallis. *Atmospheric and oceanic fluid dynamics*. Cambridge University Press, 2017.

Thomas R Parish and John J Cassano. The role of katabatic winds on the Antarctic surface wind regime. *Monthly Weather Review*, 131(2):317–333, 2003.

Christophe Genthon, Michael S Town, Delphine Six, Vincent Favier, Stefania Argentini, and Andrea Pellegrini. Meteorological atmospheric boundary layer measurements and ECMWF analyses during summer at Dome C, Antarctica. *Journal of Geophysical Research: Atmospheres*, 115(D5), 2010. doi: 10.1029/2009JD012741. Publisher: Wiley Online Library.

---

## Author Response (AR3)

**RESPONSE TO THE SECOND ROUND OF REVIEW**

**REVIEW OF DAVRINCHE ET AL., 2023 – UNDERSTANDING THE DRIVERS OF NEAR-SURFACE WINDS IN ADÉLIE LAND, EAST ANTARCTICA**

Once again, we thank the reviewers for their valuable and helpful comments on the manuscript. We propose to implement the following changes in the final version.

Black = reviewer comment / Blue = author's response / *Italic* = revised text.

**1. Reviewer #1**

Line 15: "northward" should surely be "southward"? This has been replaced: *They transport cold surface air northward, which causes warmer subpolar air masses to rise and travel southward to replenish the cold air removed* .

Advection is computed as the scalar product of wind speed and horizontal wind speed divergence." I assume that you are trying to write out the advective tendency term, (u.grad)u, in plain language? However, the words chosen don't make sense. Wind speed is a scalar, so you can't compute its divergence or form a scalar product with it. This was indeed a mistake, this has been corrected: *Advection is computed as the scalar product of the wind vector and its horizontal gradient.*

**2. Reviewer #1**

The authors' response to my suggestion regarding the vector directions on Figure 7 is still of concern to me. While I disagree with it I am willing to accept the authors' preference to show equivalent geostrophic wind direction for each force in Figure 7. But, I found the revised text that states that the katabatibc acceleration increases the wind speed in the cross-slope direction to be confusing. I agree that the katabatibc acceleration, which points in the downslope direction, will result in a geostrophic flow oriented in the cross-slope direction but the reality is that within the boundary layer an increase in the katabatibc acceleration results in increased downslope flow due to the three-way balance between the katabatibc, Coriolis and frictional acceleration. The text here needs to be revised to more clearly explain how the actual near surface flow arises from the balance of the various forces shown in Figure 7 since the equivalent geostrophic direction of each force is not what is seen in the actual winds. We thank the reviewer for the insightful comment. We have added a few sentences to explain better the rotation of the vectors: *As we are in the quasi-geostrophic stationary conditions detailed in Section 4.1, we can neglect the first temporal derivative of wind speed. Consequently, the resulting wind speed is the sum of all the equivalent geostrophic wind speeds associated to the 5 accelerations detailed in Fig.7(a-e). Therefore,*

*we show here on Fig.7 the direction of the equivalent geostrophic winds (which are rotated by 90° to the left with respect to the acceleration vectors). The same maps with the direction of the acceleration vectors are presented in Fig. S8 of the supplement.*
*Wind vectors associated with the katabatic acceleration are therefore always directed in the cross-slope direction. However, note that an increase of the katabatic acceleration does not increase the wind speed purely in the cross-slope direction because of the action of the turbulent acceleration.*

Personally, I think this is most easily done by showing the force vectors in the direction that they actually act rather than as equivalent geostrophic vectors. Since the authors' preference is to show the forces as equivalent geostrophic vectors I kindly ask that a supplemental figure be added that shows the results in Figure 7 with the vectors oriented in the direction the forces act so that readers that prefer this perspective can easily see this.

A supplemental figure (S8) has been added that shows the vectors oriented in the direction of the force.